# Structural basis of BAK activation in mitochondrial apoptosis initiation

Geetika Singh[1,2,3], Cristina D. Guibao[1,2], Jayaraman Seetharaman[1], Anup Aggarwal[1,2], Christy R. Grace[1], Dan E. McNamara[1,2], Sivaraja Vaithiyalingam[1], M. Brett Waddell[1] & Tudor Moldoveanu [1,2✉]

BCL-2 proteins regulate mitochondrial poration in apoptosis initiation. How the pore-forming BCL-2 Effector BAK is activated remains incompletely understood mechanistically. Here we investigate autoactivation and direct activation by BH3-only proteins, which cooperate to lower BAK threshold in membrane poration and apoptosis initiation. We define in trans BAK autoactivation as the asymmetric "BH3-in-groove" triggering of dormant BAK by active BAK. BAK autoactivation is mechanistically similar to direct activation. The structure of auto-activated BAK BH3-BAK complex reveals the conformational changes leading to helix α1 destabilization, which is a hallmark of BAK activation. Helix α1 is destabilized and restabilized in structures of BAK engaged by rationally designed, high-affinity activating and inactivating BID-like BH3 ligands, respectively. Altogether our data support the long-standing hit-and-run mechanism of BAK activation by transient binding of BH3-only proteins, demonstrating that BH3-induced structural changes are more important in BAK activation than BH3 ligand affinity.

[1] Department of Structural Biology, St. Jude Children's Research Hospital, Memphis, TN, USA. [2] Department of Chemical Biology and Therapeutics, St. Jude Children's Research Hospital, Memphis, TN, USA. [3] Integrative Biomedical Sciences Program, University of Tennessee Health Sciences Center, Memphis, TN 38163, USA. ✉email: tudor.moldoveanu@stjude.org

Mitochondrial poration by pore-forming BCL-2 Effectors, BAK and BAX, initiates apoptosis[1–3]. The Effectors are activated by cellular stress (e.g., DNA damage), and through poorly understood mechanisms orchestrate release of mitochondrial proteins that activate apoptotic caspases[4–7]. Active caspases dismantle cells promoting their clearance by phagocytes with minimal immunogenic consequences for the organism[8]. Blockade in mitochondrial poration by deregulation of BCL-2 family proteins is a hallmark of cancer[9]. Consequently, BCL-2 proteins are sought after drug targets, which stimulated development of prodeath drugs to overcome tumor resistance to apoptosis[10,11].

Mitochondrial poration is induced upon activation of Effectors by prodeath BCL-2 homology 3 (BH3)-only Initiators[12–24]. Poration ensues when prosurvival BCL-2 Guardians, which block poration, are outnumbered by active prodeath Initiators and Effectors[3–6,25]. The mechanism of BAK activation is incompletely understood. BAK is dormant in healthy cells[25]. Its apo structure exhibits a stable globular conformation incapable of executing poration[26]. BAK activation is triggered by a transient "hit-and-run", low-affinity binding interaction of activated BID, BIM, and possibly other BH3-only proteins at the canonical hydrophobic groove of BAK[16,18,19,27,28]. A recent report implicates BAK activation by interaction of BH3-only proteins BMF and HRK at an alternative site on the opposite face from the canonical groove[29]. How BH3 binding induces BAK to change conformation and awaken during apoptosis is unclear. Conformational changes induced upon BAK activation by BH3-only proteins involve permanent exposure of BH4 helix α1 and transient exposure of BH3 helix α2[19,25,30–32]. BIM BH3 peptide ligand complexes with BAK (and similar complexes with BAX[17,33]) have revealed cavities that form at the bottom of the activation groove at the interface with helix α1, which reflect destabilization of Effectors assumed to promote BAK and BAX activation[27]. Rationally designed inhibitory BIM BH3 peptide ligands "glued" through nonnatural amino acids-mediated salt bridges to helix α1 at the bottom of the activation groove block BAK activation and membrane poration, implicating this region in BAK activation or inactivation[27]. Antibodies (Ab) against epitopes at the N-terminus of the α1-α2 loop in BAK and BAX destabilize this region and promote Effector activation[34]. The structure of an activating Ab (aAb) bound to BAX indicated unfolding of the C-terminal turn of helix α1, further providing mechanistic support for Effector activation triggered by helix α1 destabilization[34,35]. Helix α1 destabilization promotes its unfolding and dissociation, which presumably facilitates exposure of the BH3 helix α2. Prosurvival BCL-2 Guardians bind to the exposed BH3 of active Effectors sequestering them to block apoptosis[19,25]. Additionally, active Effectors form putative symmetric BH3-in-groove dimers of the "core domain", composed of helices α2–α5, presumably upon dissociation of the core from the BH4 helix α1 and helices α6-α8 known as the "latch"[3,28,36]. The core dimers may oligomerize through latch-mediated dimerization to porate the outer mitochondrial membrane[3,7,28,30,37,38] as well as through bridging by phospholipids[39], although the mechanism of poration has not been elucidated.

Studies in cell lines depleted in known BH3-only Initiators have indicated that Effectors do not always require direct activation, suggesting possible alternative mechanisms of activation[40,41]. BAK autoactivation has recently been shown biochemically[42], but its mechanistic basis has not been elucidated. Here we show that autoactivation involves binding in trans of the BH3 helix α2 of active BAK to the activation groove of dormant BAK. We determined the crystal structure of autoactivated BAK BH3-BAK complex to reveal the molecular recognition mechanism and conformational changes in the protein core that destabilize helix α1. Moreover, we unequivocally elucidate helix α1 destabilization

in the crystal structure of directly activated BAK bound to a rationally-designed BH3 ligand, thus demonstrating a common basis for both processes. Furthermore, we find that direct activation cooperates with autoactivation in BAK-mediated poration and apoptotic response.

## Results

**BH3-in-groove asymmetric dimerization as the basis of BAK autoactivation.** Human BAK-ΔTM-His$_6$ (BAK)[26] spontaneously permeabilizes liposomes above a certain dose (>100 nM, Fig. 1a). In these dye-release assays[43–46], BAK is recruited via His-tag to liposomes containing 2.5 μM of the lipid 1,2-dioleoyl-sn-glycero-3-[(N-(5-amino-1-carboxypentyl)iminodiacetic acid) succinyl] (nickel salt), DGS-NTA(Ni). At 200 and 400 nM BAK spontaneously permeabilizes liposomes, suggesting its autoactivation; at lower BAK levels direct activation by BID BH3 is required for permeabilization (Fig. 1a, b and Supplementary Fig. 1a, b). BAK BH3 peptide weakly binds to BAK by surface plasmon resonance (SPR K$_D$ 67 μM, Supplementary Fig. 1c and Supplementary Table 1), and it activates BAK in liposome permeabilization assays similarly to BID BH3 (Fig. 1c, d).

We hypothesize that exposed BH3 region of active BAK activates dormant BAK in trans to contribute to liposome permeabilization (Fig. 1a), and we seek to elucidate this mechanism. To stabilize and crystallize autoactivated BAK, we disulfide-tethered BAK BH3GGC peptide to engineered G184C BAK (herein referred to as BAK BH3-BAK complex, Fig. 1c and Supplementary Fig. 2a). We determined the crystal structure of this complex to 3.1 Å resolution, which revealed 10 unique complexes in the asymmetric unit (Supplementary Table 2). Cu$^{2+}$ acts as crystal contact coordinating two BAK monomers (Supplementary Fig. 2b). The GGC linker is largely invisible in all 10 complexes suggesting its intrinsic disorder and minimal contribution to the structure of the complex beyond tethering (Supplementary Fig. 2c).

The BAK BH3 peptide forms a helix bound to the activation groove of BAK, similar to previous BH3-bound complexes of BAK (Fig. 1e)[19,27]. However, in marked contrast to those previous BAK complexes, autoactivated BAK exhibits rearrangements of a buried electrostatic network with helix α1 at the bottom of the activation groove. In particular, helix α1 contacts with the rest of the domain at the bottom of the activation groove are disrupted compared to apo BAK (Fig. 1f, Supplementary Fig. 2c, d and Supplementary Movie 1). In apo BAK helix α1 electrostatic network is stabilized by 2 hydrogen bonds between α1 R42 and α3 D90 and 2 hydrogen bonds between α1 E46 and α5 R137 (Fig. 1f and Supplementary Fig. 2d). Of the 10 autoactivated BAK here, seven feature 2 hydrogen bonds between α1 R42 and E46 and α2 N86; one has only 1 hydrogen bond between α1 R42 and α2 N86; and two (those with lowest b-factors) show no visible density for this network (Fig. 1f, Supplementary Fig. 2c, d and Supplementary Table 3). The destabilized conformation of the electrostatic network is unique to autoactivated BAK providing the structural basis of BAK triggering by BAK BH3. Comparison between autoactivated BAK and previous BH3-bound BAK structures will be presented later.

**Autoactivation-impaired BAK mutants porate liposomes upon direct activation.** To probe autoactivation with full-length BAK, we mutated hydrophobic residues at positions (0)−(6) of the BH3 to alanine or arginine, and D83, which forms the conserved salt bridge (s) with R127, to alanine (Fig. 2a). The BH3 residues participate in autoactivation from the ligand side [hydrophobic positions (0)–(5) and D83(s)] and also from the receptor groove side [hydrophobic positions (3)–(5)] (Fig. 2a). Thus, substitutions at positions (3)–(5) should affect receptor and ligand activities

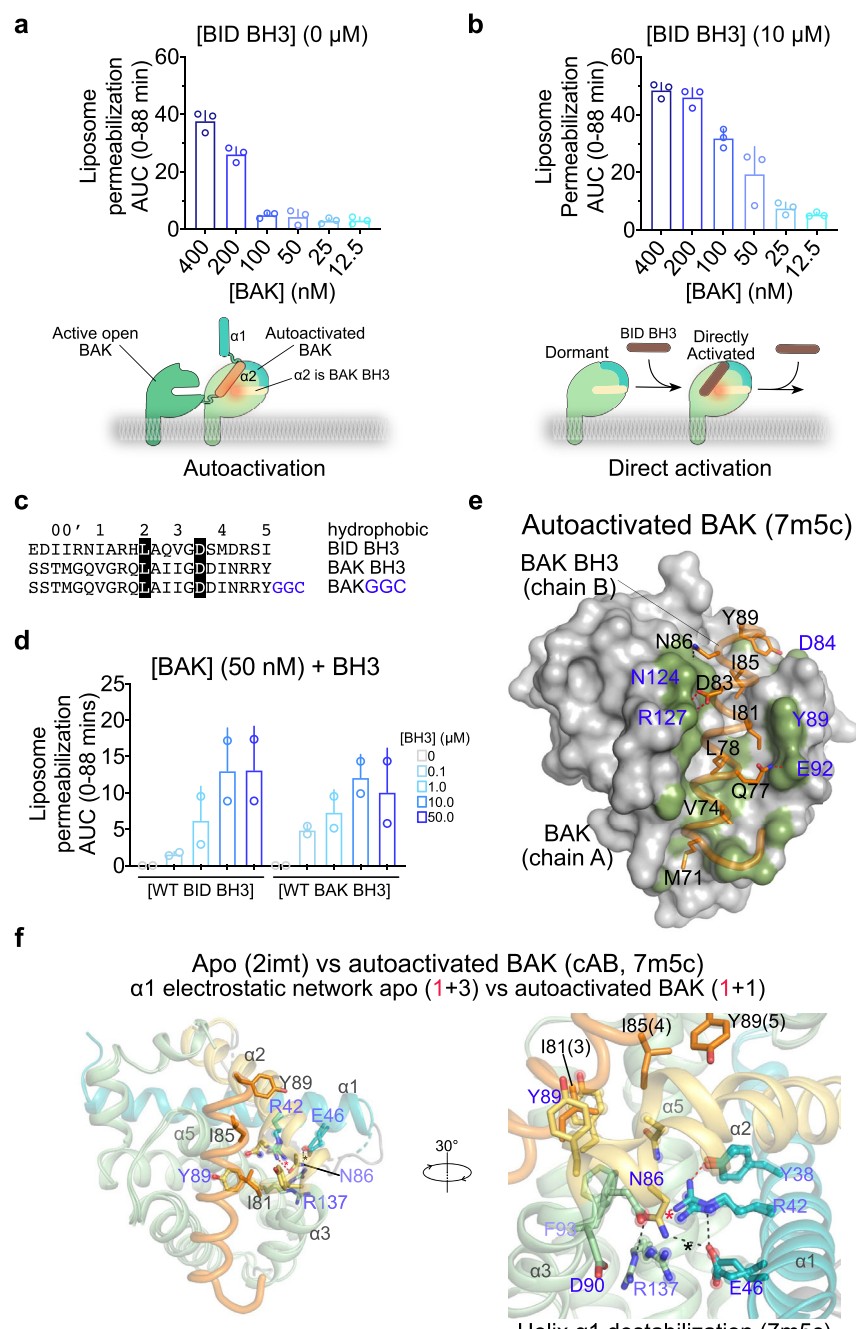

**Fig. 1 BH3-in-groove BAK autoactivation by destabilization of helix α1. a, b** BAK autoactivation and direct activation revealed in liposome permeabilization assays quantified as area under the curve (AUC) of kinetic traces in Supplementary Figs. 1 a, b. Data are presented as mean + SD from one representative of $n = 4$ experiments each of $n = 3$ technical replicates. **c** BH3 peptide sequences. Conserved positions in BH3-only Initiators and Effectors are highlighted. The GGC linker (blue) was designed for covalent tethering to BAK G184C. **d** Liposome permeabilization assays quantified as AUC of kinetic traces from Supplementary Fig. 4 for WT BAK activation by WT BID and BAK BH3 peptides. Data are presented as mean + SEM of $n = 2$ experiments each of $n = 3$ technical replicates. **e** Surface representation of BAK in complex with BAK BH3 peptide showing van der Waals contacts (≤4 Å, green). **f** Overlay of apo and autoactivated BAK reveals the mechanistic basis of autoactivation as BAK BH3 binding-induced destabilization of the electrostatic network within the protein core at the interface of helices α1, α2, α3, and α5. Apo residues are rendered as sticks and spheres and their electrostatic network contacts (dashed lines) are excluded. Hydrogen bonds ≤3.2 Å (red) and ≤3.6 Å (black) between helix α1 and the rest of the domain identified with * are summarized in brackets, Supplementary Fig. 2d and Supplementary Table 3.

and are predicted to impact BAK activation more severely than mutations of positions (0)–(2), which affect only ligand activity. Remarkably, M71A(0) and V74A(1) behave similar to WT BAK in autoactivation or direct activation in liposome permeabilization assays (Fig. 2b and Supplementary Fig. 3a), suggesting that these residues do not contribute substantially to autoactivation. In

contrast, almost all (2)–(5) and (s) missense mutants were considerably impaired in autoactivation (Fig. 2b and Supplementary Fig. 3a). The exception was I85A(4), which showed activity comparable to WT; juxtaposition of I85 from the ligand and receptor in the BAK BH3-BAK complex explains why this position can tolerate alanine but not arginine substitution. Despite

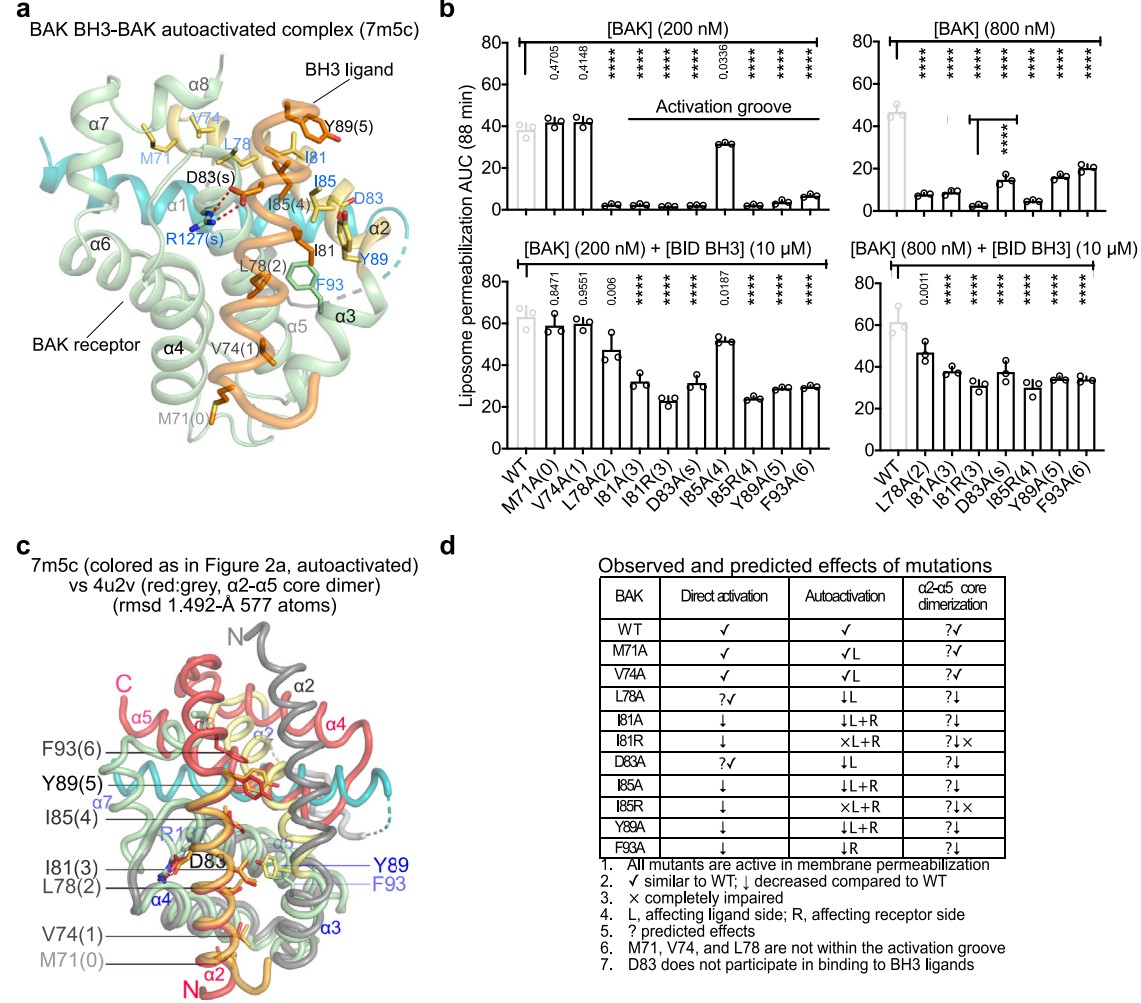

**Fig. 2 Direct activation of autoactivation-impaired BAK mutants in vitro. a** Cartoon representation of autoactivated BAK identifying residues mutated in BAK BH3 and the activation groove. **b** Liposome permeabilization assays quantified as AUC of kinetic traces from Supplementary Fig. 3a at two BAK doses reveal complete and partial impairment in autoactivation (top) and direct activation by BID BH3 peptide (bottom), respectively, for the majority of mutants. Data are presented as mean + SEM of $n = 3$ experiments each of $n = 3$ technical replicates. Adjusted $p$ values indicated above each bar were calculated by multiple comparisons using one-way ANOVA with Dunnett (200 nM BAK ± BID BH3, 800 nM BAK + BID BH3) or Tukey test (800 nM BAK); ****$P < 0.0001$. 95% confidence interval of differences are presented in the Source Data File. **c** Cartoon representation of BAK BH3-BAK structure overlaid onto that of α2–α5 core BAK dimer. **d** Summary of observed and predicted (indicated with ? prefix) effect of mutations on direct activation, asymmetric BH3-in-groove autoactivation, and symmetric BH3-in-groove α2–α5 core dimerization.

impairment in autoactivation, all missense mutants were directly activated by BID BH3 peptide, although they induced lower level of permeabilization compared to WT BAK (Fig. 2b and Supplementary Fig. 3a).

We note that asymmetric autoactivation uses the same interface as direct activation and symmetric α2-α5 core dimerization (Fig. 2c and Supplementary Fig. 3b, c); therefore, mutations in the groove are expected to affect these processes sequentially (direct activation > autoactivation > core dimerization). We investigate the effects of mutations affecting direct activation and autoactivation further here. To validate the mechanism of asymmetric BH3-in-groove autoactivation, we first investigated the ability of WT and mutant BAK BH3 peptide ligands to activate WT BAK receptor. Like full-length proteins, WT, M71A(0), and V74A(1) BAK BH3 peptides potently activate WT BAK in liposome permeabilization assays, whereas I81R(3) and I85R(4) BAK BH3 peptides are inactive (Supplementary Figs. 3d, 4a−c). Remarkably, half-maximal dye release in liposome permeabilization assays (EC50) calculated from plots of AUC against peptide dose occurred at 33–46 nM

for active peptides (Supplementary Figs. 3d, 4b), which is ~three orders of magnitude lower than the affinity of BAK for WT BAK BH3 peptide ($K_D$ 67 μM, Supplementary Table 1). This means BAK activation is very efficient even at low occupancy of BAK BH3 activating ligand.

We next compared activation of mutant BAK receptors by WT BID BH3 and WT BAK BH3 peptide ligands. At concentrations above 1 μM, both peptide ligands would activate WT or V74A(1) BAK (Supplementary Figs. 3e, 4d, e, and Supplementary Table 1). In contrast, neither peptide ligands could promote liposome permeabilization by 50 nM I81A(3), I81R(3), or D83A(s) mutant BAK receptors, which are impaired in autoactivation (Supplementary Fig. 4f), but they could promote liposome permeabilization by 200 nM D83A(s) BAK (EC50 of 4–7 μM) supporting combined contributions of both direct activation and autoactivation mechanisms to membrane permeabilization by BAK (Supplementary Fig. 3e, 4e).

To control for effect of mutations on protein stability, which may confound interpretation of impairment in BAK auto-activation, we measured the $T_m$ of BAK WT and mutants

(Supplementary Fig. 3f). We found that variations in $T_m$ among autoactivation-impaired mutants did not correlate with their effects on autoactivation. For instance, I81A(3) and D83A(s) mutations substantially decreased BAK's $T_m$ compared to WT, but other missense mutations led to increase in $T_m$, with buried hydrophobic core mutants L78A(2) and F93A(6) exhibiting the highest $T_m$ values, which may impair activation. These analyses suggest that thermal stability changes do not contribute to the effects of the mutations on autoactivation.

Altogether, our in vitro investigations support the mechanism of in trans BAK autoactivation by asymmetric BH3-in-groove dimerization as suggested by the new structure. BH3 and groove mutations impaired in autoactivation respond to direct activation albeit they are less efficient than WT BAK in permeabilizing liposomes.

**Cooperation of autoactivation and direct activation in apoptotic response.** Investigation of direct activation and autoactivation of BAK in cells containing a full repertoire of BCL-2 proteins is challenging because of heterogeneity of possible protein-protein interactions[19,25,30,47]. To overcome this limitation and demonstrate autoactivation and direct activation of BAK in cells, we used the BCL2allKO HCT116 cell line, which was previously engineered to lack expression of 17 BCL-2 family genes (BCL-2, BCL-xL, BCL-w, MCL-1, A1, BAK, BAX, BAD, BID, BIK, BIM, BMF, HRK, Noxa, PUMA, Bnip3, and Nix)[41], and stably reconstituted it with mCherry-full-length-human-BAK (mC-BAK) based on doxycycline (Dox)-inducible Tet-On 3G retroviral system (Fig. 3a and Supplementary Fig. 5a, b). We further modified these cells with BID constitutively expressed from retroviral pMX-full-length-human-BID-IRES-GFP (Fig. 3b and Supplementary Fig. 5c, d). We reasoned that Dox-induced expression fine-tunes mC-BAK levels and autoactivation (Fig. 3a), while TRAIL + cycloheximide (CHX)-induced cleavage and activation of BID promotes direct activation of mC-BAK (Fig. 3b).

We produced BCL2allKO HCT116 cell lines expressing empty vector (EV), and mC-BAK mutants V74A(1) (WT-like), I81A(3), I81R(3), and D83A(s). Immunoblotting revealed low, intermediate, and high expression levels for V74A(1), I81A(3) and D83A(s), and I81R(3) mC-BAK, respectively (Supplementary Fig. 5a). However, we could not produce mC-WT BAK in BCL2allKO HCT116 cells using the Tet-On 3G strategy. We thus used V74A(1) mutant as a surrogate for the WT BAK in these experiments, since V74A(1) and WT BAK respond similarly to apoptotic inducers actinomycin D and TRAIL + CHX when expressed constitutively as pMX-BAK-IRES-GFP in $bak^{-/-}$ $bax^{-/-}$ HCT116 cells, although we note difficulties in establishing high expression for the mutant compared to WT BAK even after four rounds of sorting GFP positive cells (Supplementary Fig. 5e, f).

We monitored Dox-induced cell death over time by IncuCyte fluorescence imaging and at the endpoint by fluorescence-activated cell sorting (FACS) analysis (see "Methods"), and observed that V74A(1) BAK induces cell death faster than I81A(3) or D83A(s) irrespective of Dox dose (Supplementary Fig. 5g, h). In contrast, I81R(3) BAK is inactive irrespective of Dox dose (Supplementary Fig. 5h). The caspase inhibitor qVD blocked cell death indicating apoptosis (Supplementary Fig. 5g). To investigate dose-dependent autoactivation in apoptotic response we compared autoactivation-impaired mutant D83A(s) to WT-like mutant V74A(1) in more detail, and performed immunoblotting analysis to control for protein expression levels. We found that D83A(s) is consistently less potent at inducing apoptosis than V74A(1), even when the latter was expressed at lower levels (Fig. 3c, d). These cellular findings are consistent with liposome permeabilization data (Fig. 2b).

To observe direct BAK activation, we monitored kinetically Dox-induced cell death of BCL2allKO HCT116 cells expressing V74A(1) or D83A(s) mC-BAK and BID by IncuCyte imaging SYTOX Green uptake. mC-BAK expression for 6 h followed by addition of TRAIL + CHX resulted in Dox dose-dependent cell death (Fig. 3e, f). BID cleavage and activation to truncated tBID was detected as early as 1.5 h after the addition of TRAIL + CHX, but it did not promote cell death in the absence of BAK (no Dox, Fig. 3e, f, Supplementary Fig. 5d). CHX blocks protein synthesis and halted production of additional mC-BAK. Compared to Dox alone, Dox + TRAIL + CHX boosted the levels of cell death induced by V74A(1) or D83A(s) at every Dox dose, suggesting a substantial contribution by direct BAK activation through TRAIL/caspase 8/BID axis (Fig. 3e, f). In all instances, qVD blocked cell death induced by Dox alone or Dox+TRAIL + CHX, indicating apoptosis. Our data suggest that autoactivation through increasing BAK levels may be further augmented through direct activation by BID in apoptotic response in the absence of other BCL-2 proteins, albeit impairment in autoactivation in D83A(s) BAK is only partially rescued through direct activation. This demonstrates cooperation of autoactivation and direct activation in triggering BAK to initiate apoptosis on a clean genetic background.

**Mapping activation hotspots in the groove of BAK through BH3 ligand mutations.** Structural changes observed in autoactivated BAK prompted us to revisit and further elucidate the mechanism of direct activation, since previous structural studies of BH3-only activators in complex with BAK[18,19,27,28] have not led to a robust unifying mechanism for BAK destabilization upon activation by BH3 ligands (see below Fig. 6, Supplementary Figs. 8, 9 and Supplementary Table 3). Previous studies were focused largely on molecular recognition between the BH3 ligand and BAK by probing the ligand, while suggesting that cavities in BAK induced by BH3 ligands are critical in activation but did not test this hypothesis.

To stabilize and structurally characterize directly activated BAK complexes, we rationally designed BID-like BH3 peptides that bind BAK with high affinity compared to WT BID BH3. We systematically mutated individual hydrophobic residues (0)–(5) of BID BH3 to larger hydrophobic residues M, F, Y, or W (Fig. 4a, b) and investigated direct BAK activation by these ligands in liposome permeabilization assays. M(3), F(3), Y(4), and W(4) substitutions showed gain in direct BAK activation compared to WT BID BH3; in contrast, W(1), Y(2), W(2), W(3) and F(4) substitutions exhibited less direct BAK activation compared to WT BID BH3; all other substitutions elicit similar BAK activation as WT BID BH3 (Fig. 4c, e and Supplementary Fig. 6a, b).

We also tested the monosubstituted peptides in competitive fluorescence polarization (FP) binding assays, which measure displacement from GST-BAK of BID BH3 stabilized alpha helix of BCL-2 proteins (SAHB) C-terminally tagged with fluorescein and thus informs on peptide binding affinity. All substitutions in the N-terminal half of the peptide decreased affinity (Supplementary Fig. 6c). In contrast, substitutions in the C-terminal half of the peptide modestly increased affinity compared to WT BID BH3 (Fig. 4d, e and Supplementary Fig. 6c).

Our functional and binding analyses suggest that: hydrophobic pockets (1) and (2) are refractory to binding large hydrophobic residues, hence shallow; all other pockets are malleable and can accommodate large hydrophobic residues; and binding to pockets (3) and (4) promotes BAK activation relative to WT BID BH3. We note that membrane permeabilization is more efficient than binding of activators (Fig. 4e), further strengthening the notion that

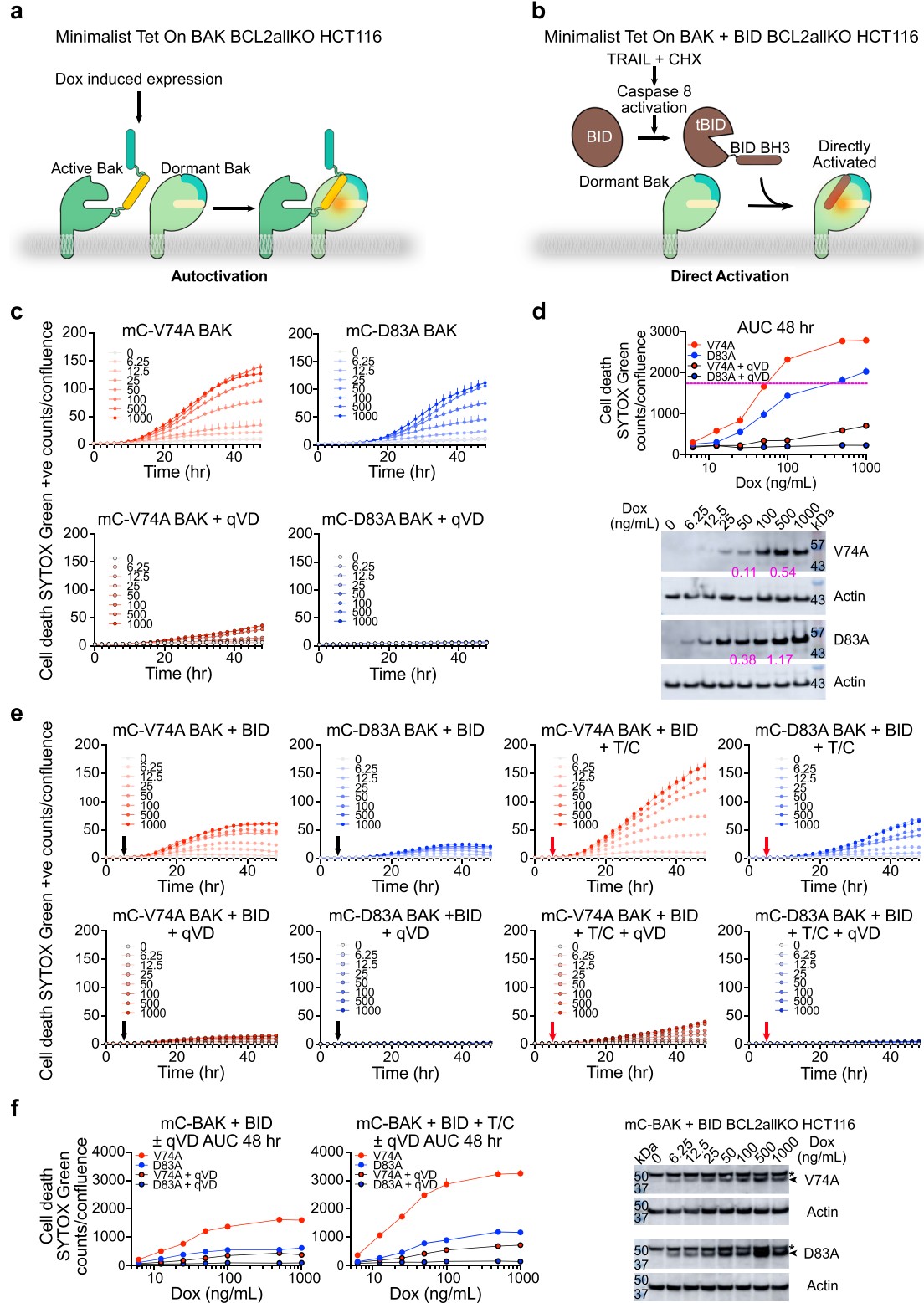

activation of a minor fraction of total BAK induces robust membrane permeabilization (Fig. 1d and Supplementary Fig. 3d, e).

**BH3 ligand-induced helix α1 destabilization as the basis of direct BAK activation**. Based on activity and binding profiles of monosubstituted peptides (Fig. 4), we produced disubstituted M(3)W(5) BID-like BH3 peptide, which exhibits ~30-fold

higher binding affinity for BAK compared to WT BID BH3, judged by ITC and SPR (Fig. 5a, b and Supplementary Fig. 7a, b). In liposome permeabilization assays this peptide is similarly potent as WT BID BH3 in direct BAK activation according to AUC against peptide dose EC50 analysis (Fig. 5a, c and Supplementary Fig. 7c).

To reveal the structural basis of direct BAK activation we determined the crystal structure of activated covalent complex

**Fig. 3 BAK autoactivation and direct activation cooperate in cells. a, b** Schematic of minimalist BAK ± BID apoptotic pathway reconstituted in BCL2allKO HCT116 cells. **c** Apoptosis of BCL2allKO HCT116 cells reconstituted with Dox-inducible V74A and D83A mCherry-BAK (mC-BAK) monitored for 48 h by IncuCyte imaging uptake of cell-impermeable dye SYTOX Green. Data are presented as mean + SD of one representative from $n = 3$ experiments each of $n = 3$ technical replicates. Dox concentration (ng/mL) is inset. **d** AUC of kinetic traces in (**c**) and representative immunoblots from $n = 2$ independent experiments. The purple line indicates similar cell death induced at ~10-fold higher expression level for D83A compared to V74A. The ratio of background-corrected mC-BAK/actin is shown in purple for 50 ng/mL and 500 ng/mL Dox. **e** Apoptosis of BCL2allKO HCT116 cells reconstituted with Dox-inducible V74A and D83A mC-BAK monitored for 48 h by IncuCyte imaging uptake of cell-impermeable dye SYTOX Green. Culture medium ± qVD (black arrows) or TRAIL + CHX (T/C) ± qVD (red arrows) were added at 6 h after addition of Dox. Data are presented as mean + SD of one representative of $n = 4$ experiments each of $n = 3$ technical replicates. Dox concentration (ng/mL) is inset. **f** AUC of kinetic traces in (**e**) and representative immunoblots from $n = 2$ independent experiments after 24 h of incubation with Dox+qVD. BAK, arrowhead; *, nonspecific band.

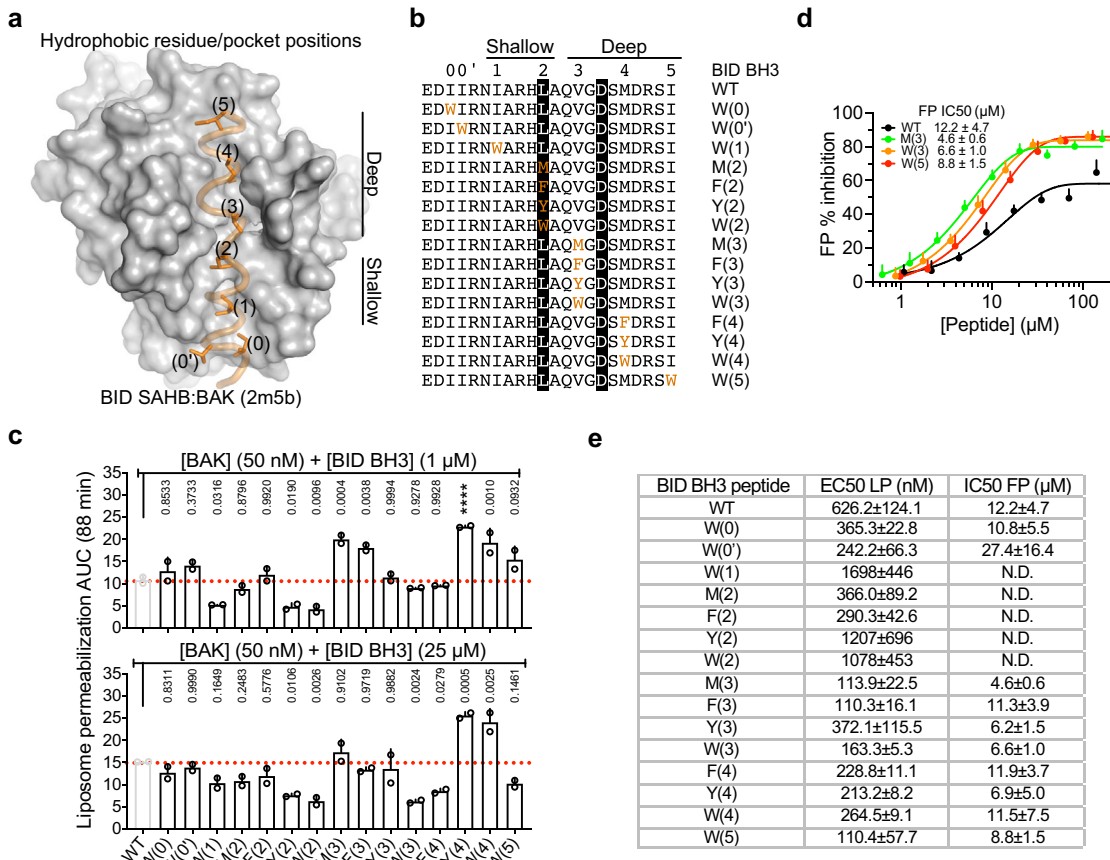

**Fig. 4 Surveying activation hotspots in the activation groove of BAK. a** Cartoon:surface representation of the BID SAHB:BAK NMR structure we determined previously showing hydrophobic residues in the BH3 helix and corresponding hydrophobic pocket in the activation groove of BAK. **b** Alignment of BID-like BH3 peptide sequences indicating M, F, Y, or W substitutions (orange). Positions conserved in BH3 regions of BCL-2 proteins are highlighted black. **c** Direct activation mode liposome permeabilization with BID BH3 peptides in (**b**) quantified by AUC of kinetic traces in Supplementary Fig. 6a. Data are presented as mean + SEM of $n = 2$ experiments each of $n = 3$ technical replicates. Adjusted $p$ values indicated above each bar were calculated by multiple comparisons to WT BAK using one-way ANOVA with Dunnett test; ****$P < 0.0001$. 95% confidence interval of differences are presented in the Source Data File. **d** Competitive fluorescence polarization (FP) assay measuring displacement of BID SAHB-fluorescein peptide from GST-BAK by BH3 peptides in (**b**). Data are presented as mean ± SD from $n = 9$ experiments. **e** Summary of liposome permeabilization (LP) EC50 values and fluorescence polarization (FP) IC50 values for the functional and binding data in panels (**c**) and (**d**) and Supplementary Fig. 6.

M(3)W(5) BID BH3GGC-G184C tevBAK [M(3)W(5) BID BH3-BAK] disulfide tethered as BAK BH3-BAK complex (Fig. 5d, Supplementary Fig. 7d and Supplementary Table 2). We observed two similar complexes in the asymmetric unit (Supplementary Fig. 7e, f), and both exhibit destabilized helix α1 electrostatic network in the protein core as observed in the BAK BH3-BAK complex (Figs. 1f, 5e; electron density shown in Supplementary Fig. 9a). Directly activated BAK exhibits 2 hydrogen bonds between α1 R42 and α2 N86 (Fig. 5e, Supplementary Table 3, and Supplementary Movie 1). The electrostatic network in the M(3) W(5) BID BH3-BAK complex resembles that observed in the

BIM-RT:BAK complex (PDB 5VWV; Fig. 6b, Supplementary Fig. 9a, c). We validated BAK destabilization upon direct activation by determining $T_m$ values of apo BAK and the M(3) W(5) BID BH3-BAK complex ± DTT. The reduced complex (15 μM, 95% peptide occupancy estimated from $K_D$) is significantly destabilized compared to apo BAK or the covalent complex (15 μM, 100% peptide occupancy based on the crystal structure, Supplementary Fig. 7h). Through rational design and structural analysis, we expand the list of activated BAK complexes, supporting requisite BH3 ligand-induced destabilization of helix α1 (see below).

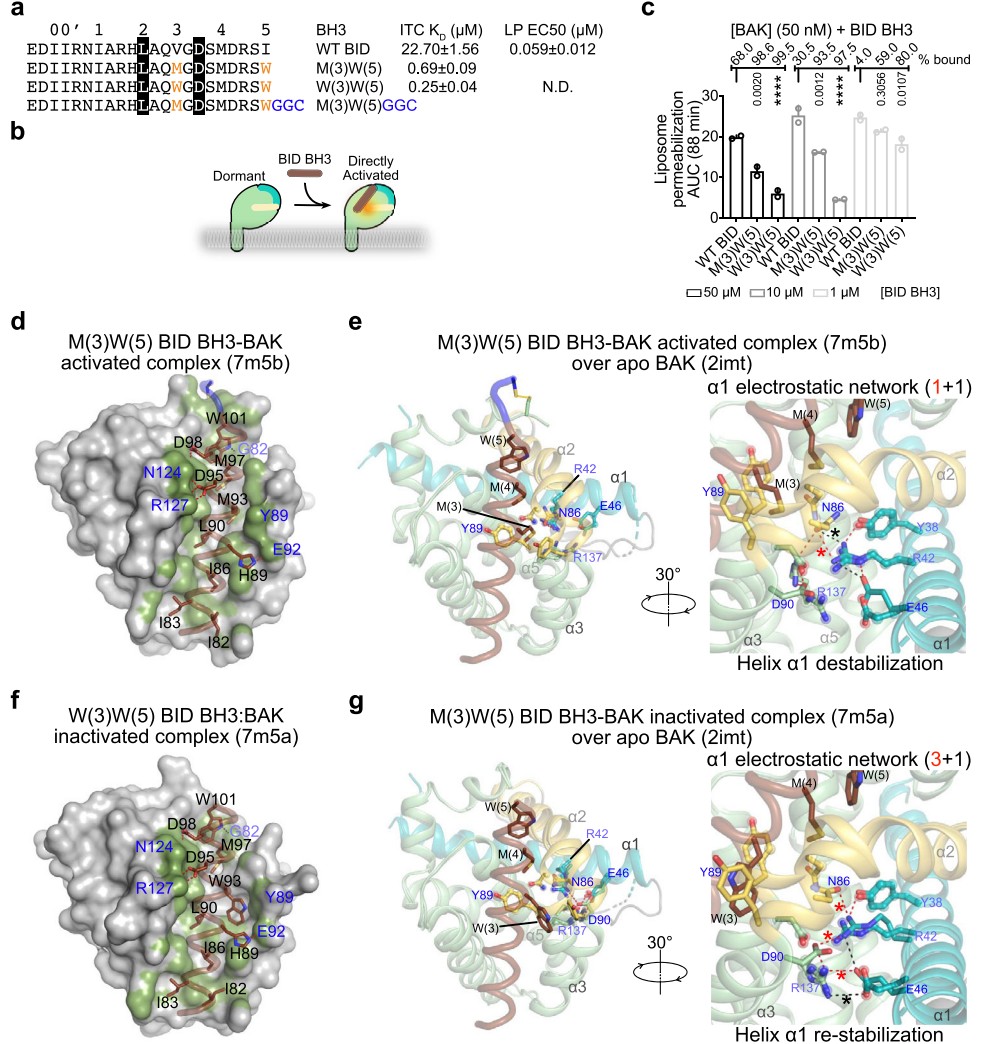

**Fig. 5 BH3 ligand binding to the activation groove regulates helix α1 stability to activate or inactivate BAK. a** BH3 peptide sequences, their binding constants determined from isothermal titration calorimetry (ITC) with BAK, and EC50 values from liposome permeabilization (LP EC50). ITC traces and binding parameters are summarized in Supplementary Fig. 7a. GGC linker is for tethering the peptide to G184C of BAK. **b** Model for direct BAK activation. **c** AUC quantification of kinetic traces for liposome permeabilization in Supplementary Fig. 7c. We estimated complex formation according to % bound = ([BID BH3] × 100)/($K_D$ + [BID BH3]). Data are presented as mean + SEM of $n = 2$ experiments each of $n = 3$ technical replicates. Adjusted p values indicated above each bar were calculated by multiple comparisons using one-way ANOVA with Tukey test; ****$P < 0.0001$. 95% confidence interval of differences are presented in the Source Data File. Cartoon:surface representation for directly activated complex (**d**) and inactivated complex (**f**) of BAK bound to BID-like BH3 peptides. Van der Waals contacts (≤4 Å) at the peptide interface are shaded green. The GGC peptide linker is colored blue (d). See also Supplementary Fig. 7 and Supplementary Table 3. **e, g** Overlay of apo and directly activated and inactivated BAK identifying key amino acids involved in helix α1 stabilization. Apo residues are rendered as sticks and spheres and their electrostatic network contacts (dashed lines) are excluded. Remarkably, the electrostatic network is destabilized and re-stabilized in these complexes, respectively, compared to that observed in apo BAK. Hydrogen bonds ≤3.2 Å (red) and ≤3.6 Å (black) between helix α1 and the rest of the domain identified with * are summarized in brackets, Supplementary Fig. 7g, i, and Supplementary Table 3.

We also designed a high-affinity disubstituted W(3)W(5) BID BH3 peptide, which activates BAK efficiently at low doses, yet inhibits BAK at high doses in liposome permeabilization (Fig. 5a–c and Supplementary Fig. 7a–c). We determined the crystal structure of the non-covalent W(3)W(5) BID BH3:BAK complex, which reveals the largest opening seen in BH3:BAK complexes induced by W(3) displacement of Y89 side chain at the center of the activation groove of BAK (Fig. 5f and Supplementary Table 2). Strikingly, the buried electrostatic network that stabilizes helix α1 in this complex is re-established as in apo BAK, involving two hydrogen bonds between α1 R42 and α2 N86 and α3 D90, and two hydrogen bonds between α2 E46 and α5 R137 (Figs. 1f, 5g, Supplementary

Table 3, and Supplementary Movie 1, electron density shown in Supplementary Fig. 9b). Our structure reveals a mechanism for BAK inactivation that differs from previous complexes of BAK inactivated by rationally-designed BIM-like BH3 ligands that directly glue helix α1[27]. The W(3)W(5) BID BH3-BAK structure bolsters the mechanistic link between the BH3 ligand binding-induced helix α1 stabilization and destabilization, and inactivation and activation of BAK, respectively.

**Unifying mechanism of BAK activation through helix α1 destabilization.** It has been hypothesized that BH3 ligand-induced cavities represent the underlying structural basis of

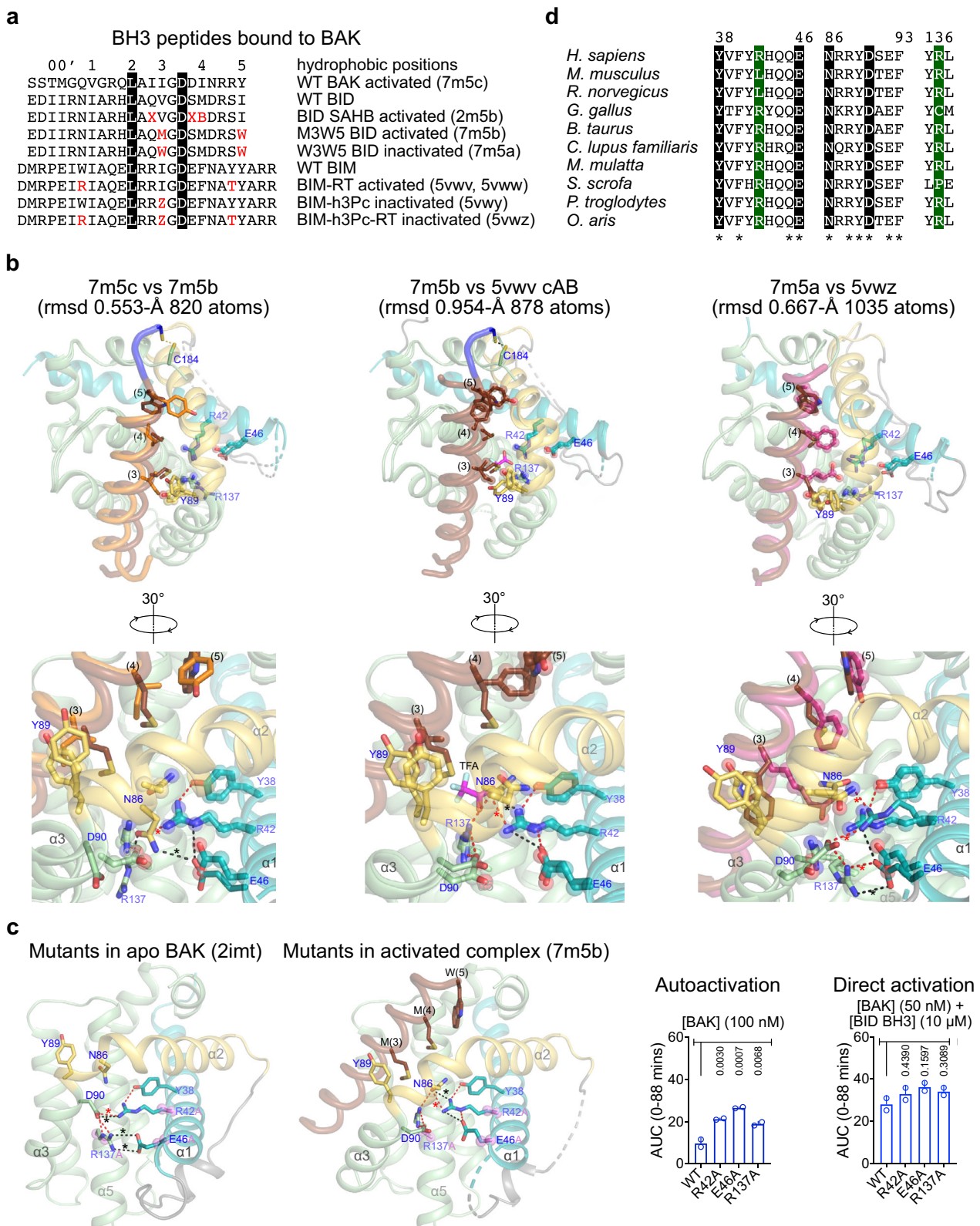

Effector activation[17,27,33]. To revisit this hypothesis considering the five available crystal structures of activated BH3-bound BAK, we aligned BAK BH3-BAK with apo BAK and with BID and BIM BH3:BAK complexes. We found that all exhibit overall very similar structures and conformational changes induced by BH3 ligands relative to apo BAK (Fig. 6a, b and Supplementary Figs. 8, 9). The most dramatic changes induced by BH3 ligands

occur at α2 and α3 helices of the activation groove, which move out upon ligand binding relative to their location in apo BAK. This movement creates deep cavities in the complexes of BAK with BIM BH3 peptides[27]. We computed cavities with CAVER Analyst[48] for crystal structures of apo BAK, activated BH3-BAK complexes, and inactivated W(3)W(5) BID BH3-BAK complex (Supplementary Fig. 8a). The BIM BH3:BAK complexes exhibit

**Fig. 6 Similarities and differences among BH3 ligand-bound BAK structures. a** Alignment of BH3 peptide ligands used to determine structures in complex with BAK. Unnatural amino acids B, norleucine; X, pentenylalanine; Z, pentenylcarboxylate. **b** Overlays of autoactivated and directly activated BAK (left, BAK BH3 orange), directly activated BAK by BID- and BIM-like BH3 (middle), and BID- or BIM-based BH3 inactivated BAK complexes (right, BIM-h3Pc-RT pink). Residues from 7m5b, 5vwv, and 5vwz in the left, middle, and right panels are rendered as sticks and spheres and their electrostatic network contacts (dashed lines) are excluded in the bottom views. **c** Cartoon representation of apo and activated BAK showing the location of alanine mutations in the buried electrostatic network. AUC quantification of kinetic traces for liposome permeabilization assays in Supplementary Fig. 8b. Data are presented as mean + SEM of $n = 3$ experiments each of $n = 3$ technical replicates. Adjusted $p$ values indicated above each bar were calculated by multiple comparisons to WT BAK using one-way ANOVA with Dunnett test. 95% confidence interval of differences are presented in the Source Data File. **d** Sequence alignment reveals conservation of the buried electrostatic network stabilizing helix α1, which involves Y38, R42, R46, N86, D90, and R137 in human BAK. Divergence observed in rodents (R42L), chicken (R137C), and pigs (R137P) is predicted to destabilize the α1 electrostatic network thereby lowering the threshold for BAK activation. Black and green highlights represent 100% and >75% conserved residues, respectively. An alternative electrostatic network stabilizes α1 in apo mouse BAK as shown in Supplementary Fig. 8c.

large cavities (above 270 Å$^3$) at the activation groove, as reported previously[27]. In contrast, the cavities in activated BAK BH3-BAK and inactivated W(3)W(5) BID BH3-BAK complexes are more modest in size (64–70 Å$^3$) whereas activated M(3)W(5) BID BH3-BAK complex exhibits no cavity at the activation groove. We note a large cavity in apo BAK at the bottom of the activation groove, which is not present in any of the complexes. Our analysis suggests that cavity induction may contribute to destabilizing BAK, but it does not constitute the unifying mechanism underlying BAK activation.

Alignments of BH3 ligand-activated and inactivated BAK complexes reveal that the most consistent changes occur within the buried electrostatic network that stabilizes helix α1 at the bottom of the groove (Fig. 6 and Supplementary Fig. 9). Compared to apo BAK this network is disrupted in activated complexes with BAK BH3 (7m5c), BID-like BH3 (2m5b, 7m5b), and BIM-like BH3 (5vwv, 5vww); re-stabilized in inactivated BAK complex with W(3)W(5) BID BH3 (7m5a); and reinforced via molecular glue stabilization in inactivated complexes with BIM BH3 (PDB 5VWY and 5VWZ; Supplementary Fig. 9). Overall, our analysis establishes the rules of engagement of the activation groove by BH3 ligands to activate and inactivate BAK.

We investigated the effects of mutations in residues involved in the electrostatic network stabilizing helix α1 observed in apo BAK. Mutations R42A, E46A, and R137A are hyperactive compared to WT in liposome permeabilization in the absence of BID BH3 (Fig. 6c and Supplementary Fig. 8b). Faster initial kinetics of liposome permeabilization in these mutants is likely caused by direct disruption of 2 out of 4 hydrogen bonds within the apo helix α1 electrostatic network. Upon activation by BID BH3, BAK WT and mutants permeabilize liposomes similarly (Fig. 6c and Supplementary Fig. 8b). The residues in the electrostatic network are not fully conserved in rodents, chicken, and pig, suggesting alternative mechanisms of helix α1 regulation in these species, but the BH3-ligand-induced helix α1 release mechanism observed for human BAK is predicted in other mammals (Fig. 6d and Supplementary Fig. 8c).

## Discussion

To initiate apoptosis BAK undergoes regulated unfolding adopting elusive membrane-associated conformations that porate mitochondria. Here we elucidate early changes in BAK conformation underlying direct activation by BID and autoactivation in trans (Fig. 7). Direct activation by BH3-only proteins has been investigated structurally yet the mechanism of BAK unfolding remains incompletely defined[3,18,19,27,28]. We show that autoactivation, a postulated but poorly characterized step in BAK activation, contributes substantially to apoptotic response. Autoactivation cooperates with direct activation to amplify signaling and lower the threshold of BAK required for mitochondrial poration (Supplementary Movie 2). Autoactivation is mechanistically similar to direct activation: binding of exposed BH3

of BAK or activators to the activation groove induces conformational changes in the electrostatic network at the bottom of this groove destabilizing helix α1 to initiate BAK unfolding (Fig. 7 and Supplementary Movie 1). While our reconstituted in vitro and cellular systems with BAK and its activator BID rely on increasing BAK concentration, which has not been reported as a general mechanism of apoptosis initiation, they offer the means of directly investigating BAK in the absence of other BCL-2 proteins (Fig. 7 and Supplementary Movie 2).

In cells with complex repertoires of BCL-2 proteins, BAK activation is also regulated through inhibition by the prosurvival BCL-2 Guardians, which act by sequestration of Initiators (MODE 1) or Effectors (MODE 2)[25]. Our model may explain the effective activation of dormant BAK upon derepression of MODE 2 (i.e., when active BAK is freed from complexes with prosurvival BCL-2 Guardians by BH3 mimetics); presumably this exposes BAK BH3 which can autoactivate in trans nearby dormant BAK. Autoactivation could explain Effector activation in HCT116 cells depleted in BH3-only Initiators when the prosurvival BCL-2 Guardians are antagonized, although it was speculated that Effectors are activated through a membrane permissive model (i.e., membranes activate Effectors)[40,41]. The autoactivation mechanism does not exclude this model. Our understanding of the dynamics of BAK interaction with membranes is incompletely understood, and therefore we cannot explain the initiating event leading to BAK autoactivation in vitro or in cells in the absence of BH3-only activators (Fig. 7).

Mechanistically BAK activation by BH3 ligands is incompletely defined. Published BH3 bound-BAK complexes have not unequivocally revealed the molecular basis of helix α1 destabilization upon activation[19,27]. In these studies, and similar studies with BAX, ligand-induced cavities have been associated with Effector destabilization upon activation by BH3 ligands, but their role has not been formally investigated[17,27,33]. Our calculations suggest that cavity induction at the activation groove does not correlate with BAK activation (or inactivation). Instead, our comparative analyses of activated BAK complexes support destabilization of electrostatic contacts to helix α1 at the bottom of the activation groove as the common mechanism of activation by BH3 ligands. Our new structures (10 asymmetric unit BAK BH3-BAK complexes and 2 asymmetric unit M(3)W(5) BID BH3-BAK complexes) capture destabilization of helix α1 contacts to consolidate this mechanism as the basis of BAK activation by BH3 ligands. We refer to the region of BAK from the N terminus to the end of helix α2 as the N-bundle, and to that from helix α3 to the C terminus of the globular domain as the C-bundle[19,26]. The binding of activating BH3 ligands disrupts helix α1 contacts to helices α3 and α5 of the C-bundle observed in apo BAK. BH3 ligand-activated BAK exhibits electrostatic contacts between helix α1 and helix α2 of the N-bundle, which is dynamic and susceptible to m-calpain proteolysis in endogenous mitochondrial BAK[19,25,49]

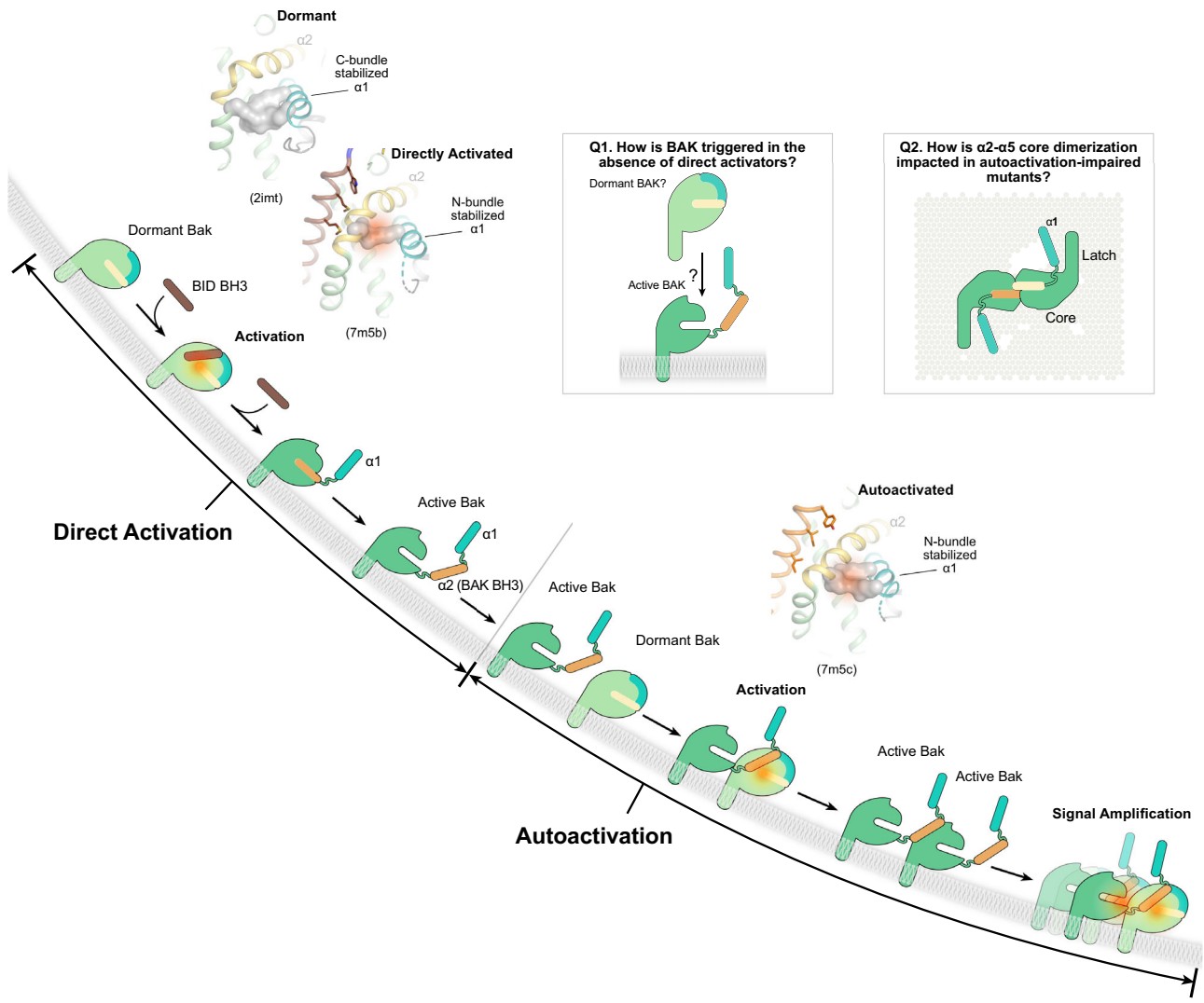

**Fig. 7 Cooperation of autoactivation and direct activation in apoptosis initiation by BAK.** Direct BAK activation by BH3-only proteins cooperates with BAK autoactivation in trans to lower BAK threshold and to amplify the response leading to mitochondrial poration (Supplementary Movie 2). BAK autoactivation involves binding the exposed BH3 of active BAK to the activation groove of dormant BAK. Our structural analyses of autoactivated (7m5c) and directly activated (7m5b) BAK reveal BH3 ligand-induced conformational changes in the protein core that promote helix α1 destabilization as the mechanistic basis of BAK activation (inset cartoons). In activated BAK complexes helix α1 electrostatic network is destabilized as it rearranges within the N-bundle (helices α1-α2), which dissociates in the presence of membranes but not in solution. In contrast, in dormant, apo BAK (2ims) and an inactivated BAK complex bound to a rationally-designed BID-like BH3 peptide (7m5a), helix α1 is stabilized by the electrostatic network within the C-bundle (helices α3-α8), which prevents N-bundle dissociation at membranes (Supplementary Movie 1). How active BAK porates membranes is unclear but oligomerization of the BAK α2-α5 core dimers has been postulated. Mechanistic questions not addressed in our study are inset.

(Fig. 7 inset and Supplementary Movie 1). NMR and hydrogen-deuterium exchange mass spectrometry experiments showed that residues 1–65 are intrinsically disordered in BH3 ligand-activated BAK[31,32]. Corroborating this mechanism, the $T_m$ of activated BH3-BAK complexes is lower than that of apo BAK, and auto-activation proceeds at lower doses of protein with mutants that disrupt the electrostatic network stabilizing helix α1 in apo BAK. Intriguingly and contrary to published results implicating R137A in impaired bridging of α2-α5 core dimers by phospholipids[39], in our hands this mutant exhibits in vitro autoactivation at lower protein dosing compared to WT and it performs like WT when activated by BID BH3. The role of helix α1 in regulating BAK activation derives also from analysis of inactivated W(3)W(5) BID BH3-BAK complex, which exhibits helix α1 stabilization resembling that in apo BAK[26] (Supplementary Movie 1). Others rationally designed inhibitory BIM-like BH3 peptides with non-

natural carboxylic acid residues at position (3), which directly contact and "glue" helix α1 to inactivate BAK[27]. The wide opening of the hydrophobic groove in W(3)W5 BID BH3-BAK complex offers a novel strategy of inactivating BAK by blocking the activation groove while stabilizing helix α1 through indirect contacts. A caveat to this strategy is that an inactivator may activate BAK if it falls off and BAK reverts to a transient intermediary state resembling the BH3-activated conformation with a destabilized helix α1 electrostatic network (Supplementary Movie 1). Our data support the long standing hit-and-run model governing BAK activation by weak affinity BH3 ligands[16], mechanistically implicating efficient BH3 ligand-induced helix α1 destabilization and BAK unfolding at membranes.

Activation of BAK, BAX, and BOK implicates helix α1 destabilization[3]. BAX activation is initiated when BH3 activators including BAX BH3 engage the noncanonical trigger site

(helices α1 and α6 and the α1–α2 loop). This promotes allosteric changes that release the TM helix α9 facilitating BAX relocalization from cytosol to mitochondria[3,20,50–52]. At mitochondria, BAX follows a similar activation path as described for BAK (Fig. 7)[3]. Asymmetric BH3-in-groove autoactivation of BAX is predicted[42]. Structural analysis suggested destabilization through BH3-induced cavity formation in autoactivated BAX, yet its consequences in membrane poration and apoptosis initiation have not been formally evaluated[17]. BOK bypasses activation by BH3 ligands, intrinsic instability driving its autoactivation; helix α1 is destabilized in BOK by a central glycine whose mutation to alanine blocks mitochondrial poration and apoptosis[43,53]. Therefore, activation of Effectors shares common features of regulated unfolding.

Elegant studies using BAK aAbs support autoactivation in trans: aAbs activate WT BAK which in turn activates BAK mutants resistant to activation by aAbs, presumably through BH3-in-groove triggering[34,42]. Our BAK BH3-BAK structure provides atomic insights into this process. Mutants in the BAK BH3 similar to ours were originally shown to be impaired in apoptosis when expressed in $bak^{-/-}$ $bax^{-/-}$ mouse embryonic fibroblasts by inhibiting BH3-in-groove dimerization[30], which was later interpreted structurally as the symmetric α2–α5 core dimers[28]. We propose that asymmetric BH3-in-groove dimerization precedes symmetric α2–α5 core dimerization. Thus, mutations in the BH3 or groove should impact autoactivation before dimerization. A caveat to this approach is that overlap of structural elements mediating BAK activation and dimerization makes it difficult to uncouple these mechanisms through mutagenesis. Thus, the cumulative effect of mutations in these elements is diminished autoactivation, dimerization, poration, and apoptotic response, and these mutations may impair also direct activation.

BAK forms symmetric BH3-in-groove α2–α5 core dimers considered as the building blocks of oligomers that porate the mitochondrial outer membrane, although the mechanism of poration is unknown[3,17,19,25,28,30,34,36–39,54,55]. Given the similarity of asymmetric BH3-in-groove (this study) and symmetric α2–α5 core dimerization[28,36,39], we find it remarkable that mutations in the BAK BH3 and the activation groove completely abolish autoactivation but not poration in vitro [e.g., I81R(3)]. This suggests that the putative symmetric BH3-in-groove dimers are either extremely tolerant towards these mutations or that alternative mechanisms of poration exist. The impact of these mutations on BH3-in-groove α2–α5 core dimerization will be examined elsewhere (Fig. 7), but it may explain the requirement for higher doses of mutant proteins to permeabilize liposomes compared to WT BAK, and similarly elevated protein levels in cells [e.g., D83A(s) vs V74A(1) BAK]. In conclusion, our study supports cooperation between BAK autoactivation and direct activation in mitochondrial apoptosis initiation revealing the unifying basis for the hit-and-run mechanism as the BH3 ligand-induced destabilization of BAK helix α1.

## Methods

**Cloning and recombinant protein expression and purification.** For structural studies, we cloned the cDNA of C terminus-truncated human MEAS-BAK[26] without the C-terminal histidine tag into pNIC28-Bsa4 vector[56], and introduced a TEV site after residue 20. Amplification and mutagenesis primers are included in Supplementary Table 4. We expressed the construct in *E. coli* T7 Express pLac I/Y cells (New England Biolabs). We grew bacteria to OD 0.8–1.0 in Luria-Bertani medium at 37 °C and induced protein expression O/N with 100 mg/L IPTG at 21 °C. We lysed bacteria by passing them through Avestin EmulsiFlex-C3. After centrifugation (16000 × *g*) to separate insoluble membranes, we batch-purified the protein on Ni$^{2+}$-NTA affinity chromatography, and S200HR size exclusion chromatography using AKTA Pure (GE Healthcare). To generate tevBAK (residues 22–186), we cleaved off the histidine tag with TEV protease, which we further purified by SPFF cation exchange chromatography (GE Healthcare). TevBAK

was >95% pure, judged by sodium dodecylsulfate polyacrylamide gel electrophoresis (SDS-PAGE). We produced G184C C166S tevBAK for structural analysis in a similar manner.

For functional studies, we made the C-terminus truncated recombinant human BAK containing C-terminal 6× histidine tag (MEAS-BAK)[19,26]. We expressed WT and BAK mutants in *E. coli* T7 Express pLac I/Y cells, and purified them similar to tevBAK, without the TEV protease digestion step. Proteins were >95% pure, judged by SDS-PAGE. Intact mass spectrometry confirmed the molecular masses of WT and mutant MEAS-BAK proteins (Center for Proteomics and Metabolomics at the St. Jude Hartwell Center for Bioinformatics).

We expressed GST-BAK, used for competitive fluorescence polarization assays, from the pRL296 vector (a gift from Mirek Cygler), batch affinity purified it on glutathione agarose resin (Goldbio), and enriched it further by S200HR size exclusion chromatography.

We exchanged the buffers of all proteins to 20 mM HEPES (pH 6.8) or 20 mM Phosphate (pH 6.8) ± DTT as needed, through several rounds of concentration and dilution by centrifugation (4000 × *g*) in Amicon centrifugal filters (Millipore). We flash froze proteins in liquid nitrogen and stored them at −80 °C until needed.

**Site directed mutagenesis.** QuikChange II XL kit (Agilent Technologies) was used to introduce mutations in BAK. Mutagenesis primers are included in Supplementary Table 4.

**Peptide synthesis.** The Peptide Synthesis facility at the St. Jude Hartwell Center for Biotechnology produced the BH3 peptides by standard FMOC-based chemistry on a polystyrene resin using the SymphonyX or the LibertyBlue Microwave Peptide Synthesizer. N-terminal acetylation and C-terminal amidation were done during synthesis. The BID BH3 stabilize alpha helix of a BCL-2 protein (SAHB)-fluoresceine (SAHB-F) peptide was acetylated at the N terminus and derivatized with ethylene diamine-fluorescene (*) at the C terminus (EDIIRNIARH-LAXVGDXBDRSIPPG*; X, (S)-2-(4-pentenyl)alanine; B, norleucine). Chemical stapling of the SAHB was performed using ruthenium-catalyzed ring closer metathesis after cleavage[19]. All peptides were HPLC purified and were >95% pure according to quality control Waters Alliance High Performance Liquid Chromatography (HPLC). Matrix-Assisted Laser Desorption Ionization Time of Flight (MALDI-TOF) mass spectrometry was used to confirm all peptides.

**Covalent conjugation of BAK and BH3 peptides.** We prepared BID or BAK BH3GGC-G184C C166S tevBAK conjugates for structure determination and thermal shift assays (TSA) by treating 10 mg/mL protein (~500 μM) with 2.5× to 10× molar excess M(3)W(5) BID or BAK BH3 peptides containing the C-terminal GGC linker in 20 mM HEPES buffer (pH 7.5–8.0). We incubated these mixtures for 24–72 h at RT. We removed excess free peptides by MonoS cation exchange or S200GL size exclusion chromatography. Purity was visually estimated by SDS-PAGE (see Supplementary information for gels).

**X-ray crystallography.** We incubated recombinant human WT tevBAK (10 mg/mL) with 500 μM CuSO4 for 2–3 h, and added 4× molar excess of W(3)W(5) BID BH3 peptide before crystallization. Crystals grew in 0.2 M potassium sodium tartrate and 20% (w/v) PEG 3350. We cryo-protected crystals in 1:1 paratone:paraffin oil mixture and stored them in liquid nitrogen. We collected X-ray data at the SERCAT 22ID beamline. We incubated the purified complex BID M(3)W(5) BID BH3GGC-G184C C166S tevBAK with 500 μM CuSO4 before crystallization. Crystals grew in 15% PEG 4000, 0.2 M NaCl, 0.1 MES pH 6.5. We cryo-protected crystals in 1:1 paratone: paraffin oil mixture and stored them in liquid nitrogen. We collected X-ray data at the SERCAT 22ID beamline. We processed diffraction images in HKL2000[57], scaled and prepared the data in CCP4[58,59], and solved the structure by molecular replacement in PHENIX[60] using published BAK structures, including apo BAK (2imt) and the BIMP$_C$: BAK complex (5vwz), as template for the M(3)W(5) BID BH3GGC-G184C BAK structure, and used this structure as template for solving the W(3)W(5) BID BH3:BAK complex. We refined the structures in PHENIX[60] and REFMAC[61] and built the model in COOT[62].

We incubated the purified complex WT BAKGGC-G184C C166 tevBAK with 500 μM CuSO4 before crystallization. Thin rod crystals (6–8 μm) grew in 0.1 M MES pH 6.5, 0.5 M Ammonium Sulfate. Several attempts of optimization did not yield better crystals. We cryo-protected crystals with 25% glycerol in mother liquor and stored them in liquid nitrogen. We collected X-ray data at the FMX beamline at the National Light Source II, Brookhaven National Laboratories. We processed diffraction images with XDS[63] and solved the structure by molecular replacement with PHENIX[60] using the M(3)W(5) BID BH3GGC-BAK complex as template. We refined the structure using PHENIX[60] and REFMAC[60] and built the model in COOT[62]. We present data collection and refinement statistics for all structures in Supplementary Table 2. Cavities at the activation groove of BAK for apo BAK, and activated and inactivated BH3:BAK complexes were estimated using CAVER Analyst 2.0[48].

**Competitive fluorescence polarization assay.** We carried out fluorescence polarization assays, measuring the displacement of the fluorescent peptide BID BH3 SAHB-F from GST-BAK, in 384-well Greiner black polystyrene plates with

flat bottom (Sigma-Aldrich). Master mix consisted of 50 mM Tris-HCl pH 7.5, 50 mM KCl, 0.1 mg/mL BSA, 2 mM DTT, 0.005% Tween-20, 10 nM BID BH3 SAHB-F, and 1 μM GST-BAK. We pin transferred peptides using Biomek FX (Beckman Coulter Life Sciences, IN) to achieve final concentration of 500−0.2 μM in 20 μL final volume. We gently centrifuged plates to remove air bubbles for 1 min at 1000 rpm, and incubated them for 60 min at RT. We measured polarization intensity in milipolarization units (mP) using a Clariostar microplate reader (BMG LABTECH) set with excitation and emission wavelength of 485 and 535 nm, respectively. Free peptide controls in each plate consisted of only BID BH3 SAHB-F and bound peptide controls containing BID BH3 SAHB-F + GST-BAK. We derived Z'-values using the equation $[1 − 3(stdev^L + stdev^H)]/(average^H − average^L)$, where $L$ is the free peptide control and $H$ is the bound peptide control[64]. Z'-values were > 0.8, which suggested excellent assay performance. We processed and visualized data using the software Robust Investigation of Screening Experiments (RISE) developed in house in the Pipeline Pilot platform (Accelrys, v.8.5.0). We performed at least three independent dose-response experiments for each peptide. We estimated $IC_{50}$ values in RISE and GraphPad Prism v.8.0a.

**Liposome permeabilization assay.** We performed liposome permeabilization assays as previously described[43–46]. Briefly, we prepared lipid films using lipid ratios similar to mitochondrial membrane composition; 40.9% phosphatidylcholine, 26.6% phosphatidylethanolamine, 9.1% phosphatidylinositol, 8.3% phosphatidylserine, 7% cardiolipin and 8.0% $Ni^{2+}$-affinity lipid 1,2-dioleoyl-sn-glycero-3-[(N-(5-amino-1-carboxypentyl)iminodiacetic acid)succinyl] (nickel salt) [DGS NTA(Ni), 2.5 μM final concentration in each assay] (Avanti Polar Lipids). We prepared 12.5 mM fluorophore; 8-aminonaphthalene-1, 3, 6-trisulfonic acid, disodium salt (ANTS) and 45 mM quencher; p-xylene-bis-pyridiniumbromide (DPX) in LUV buffer (10 mM HEPES pH 6.8, 200 mM KCl and 5 mM $MgCl_2$). We mixed the lipid film in ANTS/DPX solution by vortexing and sonication (~15 min) followed by extrusion through polycarbonate membrane of 2 μm pore size (Avanti Polar lipids) to generate homogenous large unilamellar vesicles (LUVs). We separated LUVs from free ANTS/DPX by S500 size exclusion chromatography and stored them at 4 °C in the dark. We prepared peptide and protein dilutions in 96-well black flat bottom plates (Costar) kept on ice. ANTS release from permeabilized liposomes causes an increase in fluorescence, which we monitored at 37 °C for 90 min using a CLARIOSTAR microplate reader (BMG Labtech) set at excitation and emission wavelengths of 360 and 530 nm, respectively. We normalized the data relative to maximum and minimum fluorescence induced by 3% CHAPS and buffer controls, respectively. We quantified liposome permabilization by integrating area under curve (AUC) of normalized traces based on the Simpson's equation, and analyzed that data in Excel and GraphPad Prism. AUC measurements reflect the kinetic and extent of liposome permeabilization and provide quantitative means for comparison between different experimental conditions (e.g., BAK mutants with gross or partial variation in activity). Variability in permeabilization of different liposomes is common and our analyses compare among conditions investigated with the same liposomes. Although the absolute permeabilization values may differ, the trends for the same conditions are always very similar between different liposomes.

**Isothermal titration calorimetry.** Thermodynamic binding parameters for the binding of various BID BH3 peptides to WT and mutant tevBAK were measured using a MicroCal auto-iTC 200 (Malvern Instruments). Protein samples were exchanged into 20 mM HEPES, 100 mM NaCl and 1 mM TCEP prior to the experiment. Titrations were performed by first injecting 0.5 μl of 250 or 500 μM BID BH3 peptides into a solution of 20 or 50 μM BAK followed by additional 2 μl injections. Experiments were carried out at 25 °C. Results were analyzed using Origin software (OriginLab) provided by MicroCal. Binding constants ($K_D$) and thermodynamic parameters were calculated from the average of two or three individual titrations by fitting the data to a single site binding model using a nonlinear least-squares fitting algorithm.

**Surface plasmon resonance.** We conducted SPR experiments at 20 °C using a Pioneer FE optical biosensor (ForteBio) amd histidine-tagged MEAS-BAK-$His_6$ immobilized on polycarboxylate hydrogel-coated gold chips preimmobilized with nitrilotriacetic acid (HisCap chips; ForteBio). We primed the chip in chelating buffer (10 mM HEPES pH 7.4, 150 mM NaCl, 50 μM EDTA, 0.005% Tween20) and preconditioned it at 10 μL/min with three 60 s injections of wash buffer (10 mM HEPES pH 8.3, 150 mM NaCl, 350 mM EDTA, 0.05% Tween20) and one 60 s injection of chelating buffer before charging with a 60 s injection of 500 μM $NiCl_2$ in chelating buffer. After priming into binding buffer (20 mM HEPES pH 8.0, 200 mM NaCl, 1 mM TCEP, 0.01% Tween20), we injected BAK to achieve ~500 RU of captured protein. As reference cell, we charged one flow cell on the chip with $Ni^{2+}$ without adding protein. We prepared peptides in binding buffer as a three-fold dilution series at a maximum concentration of 100 μM for M(3)M(5) and WT BID peptides and of 10 μM for all other peptides and injected them in triplicate for each concentration at a flow rate of 75 μL/min. We included a series of buffer-only (blank) injections throughout the experiment to account for instrumental noise. The peptides fully dissociated from the protein surfaces, eliminating the need for a regeneration step. We processed, double-referenced, solvent corrected, and analyzed the data using the

software package Qdat (version 4.3.1.2, ForteBio). We determined kinetic rate constants and affinities by fitting the data to a 1:1 interaction model.

**Thermal shift assay.** We carried out TSA in 20 μL reactions in 384-well PCR plate (Cat# AB1384, Thermo Fisher Scientific) in 20 mM HEPES pH 7.0, 150 mM NaCl, 2 mM DTT, SYPRO $^{TM}$ Orange (5×) and 0.25 mg/mL purified BAK. To remove air bubbles we sealed assay plates with MicroAmp Optical Adhesive, incubated them for 2–5 min, and centrifuged them (1000 × rpm). We subjected the assay plates to 25–99 °C thermal gradient with 1°/minute increment and recorded fluorescence using Applied Biosystems Quant Studio 5. We performed three experiments in quadruplicate for each sample. We performed data processing and visualization in RISE, and final data analysis in Excel and GraphPad Prism. We also carried out dye-free TSA using capillary-based Prometheus NT.48 Nano Differential Scanning Fluorimetry (Nanotemper) in the same buffer conditions with 15 μM BAK. Melting temperature ($T_m$) values were obtained from the first derivative of melting scans.

**Cloning and production of cells expressing minimalist BCL-2 family repertoire.** For mammalian cell culture experiments, we cloned the full length human BAK cDNA into pCR 2.1 TOPO vector, mutated BAK as needed, and PCR amplified the BAK gene for cloning into the BamHI and NotI sites of pRETROX mCherry vector for retroviral transduction. Amplification and mutagenesis primers are included in Supplementary Table 4. To test direct BAK activation and BAK auto-activation in cells, we produced cells stably expressing mCherry-human BAK (mC-BAK) ± human BID (BID) on a genetic background devoid of the entire BCL-2 family repertoire (all 17 BCL-2 genes), BCL2allKO HCT116, as described by the Xu Luo laboratory[41]. We expressed WT and mutant mC-BAK using the doxycycline (Dox)-based inducible Tet-On 3G retroviral system from the pRetroX-TRE3G-Puro$^R$ in BCL2allKO HCT116 cells expressing the Tet3G from pRetroX-Tet3G-Blast$^R$. We selected cells with puromycin (2.5 μg/mL) and blasticidin (2 μg/mL) starting at 2 days after retroviral transduction. To enrich for populations of cells that express mC-BAK, we treated cells with low Dox dose (25 ng/mL) and fluorescence-activated cell sorted (FACS) the mCherry positive cells. We performed the enrichment twice before cell death analysis. One caveat of our cellular system is tagging BAK with N-terminal mCherry, which may affect folding efficiency, degradation, and BAK interactome.

We achieved stable, constitutive expression of human BID in mC-BAK expressing BCL2allKO HCT116 by retroviral transduction based on pMX-human-BID-IRES-GFP. Amplification primers are included in Supplementary Table 4. We performed two rounds of FACS sorting gating on GFP-positive cells to enrich for populations of cells that express BID.

We stably reconstituted constitutively expressed BAK WT and V74A(1) using pMX-human-BAK-IRES-GFP in $bak^{−/−}$ $bax^{−/−}$ HCT116 cells. We enriched for GFP positive cell populations by up to four rounds of sorting. Cells were used in apoptosis assays.

**Cell death analysis.** We seeded cells in 24-well (100,000 cells/well), 96-well (15,000 cells/well), or 384-well (10,000 cells/well) plates coated with 10 μg/mL fibronectin. After ≥24 h of attachment and recovery, we induced cell death with Dox (up to 1000 ng/mL) and monitored SYTOX Green (25 nM) uptake by Incucyte imaging. This dye marks cells with compromised plasma membrane integrity. mC-BAK expressed before 24 h, which coincided with detectable cell death. We monitored Incucyte imaging for up to 48 h, followed by endpoint FACS analysis to measure the extent of cell death based on SYTOX Green positive cells on FACS-Calibur systems (BD Biosciences). As expected, cells with low mCherry fluorescence were refractory to cell death.

For coexpression of mC-BAK + BID, we treated cells with different doses of Dox for 24 h to induce expression of mC-BAK. At 6 h, we treated cells with TRAIL (200 ng/mL) + cycloheximide (CHX, 0.5 μg/mL) to promote BID activation under conditions of protein synthesis inhibition by CHX, and continued to record Incucyte imaging of SYTOX Green uptake up to 48 h. We blocked caspase activity with qVD (40 μM).

**Immunoblot analysis.** We monitored mC-BAK and BID expression, and BID processing to tBID by immunoblotting with monoclonal antibodies against mCherry (1:1000 dilution, 16D7, Invitrogen), BAK (1:1000 dilution, Ab-1, Calbiochem, or 3814s, Cell Signaling) and BID (1:1000 dilution, 5C9, Santa Cruz). As loading controls, we monitored global protein levels by ponceau S staining (Millipore) and actin (1:1000 dilution, C4, Millipore). We used the secondary antibodies goat anti rat IgG (1:2000 dilution, GE Healthcare) and sheep anti mouse IgG (1:2000 dilution, GE Healthcare).

**Statistical analysis.** For individual experiments, we plotted data as averages of triplicate or quadruplicate measurements with error bars of standard deviation (SD). We represented data combined from two to three independent experiments as the average and standard error of the mean (SEM). We performed statistical analyses as summarized in each Figure by one-way ANOVA using the Dunnett or Tukey methods for multiple comparisons with multiplicity adjusted, family-wise

0.05 confidence level in Prism. *P*-values are explicitly indicated above bar graphs ($P < 0.0001$; ****).

**Unique biological materials**. All unique materials are readily available from the authors. The HCT116 BCL2allKO cell line was obtained from Dr. Xu Luo at University of Nebraska through material transfer agreement.

**Reporting summary**. Further information on research design is available in the Nature Research Reporting Summary linked to this article.

## Data availability

The data that support this study are available from the corresponding author upon reasonable request. Structural data have been deposited to wwPDB.org under accession codes 7M5A, 7M5B, 7M5C. Source data are provided with this paper.

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

## Acknowledgements

We thank Ines Chen, Sebastian Ruehl, Douglas Green, Richard Kriwacki, and Babis Kalodimos of St. Jude for insightful discussions and suggestions; Xu Luo of University of Nebraska for sharing the BCL-2 allKO HCT116 cells; Sebastian Ruehl for sharing the BID cDNA; Jaeki Min of St. Jude for help with extracting fluorescence polarization raw data from RISE; Patrick Rodrigues and his team at St. Jude Hartwell Center for Biotechnology for synthesizing BH3 peptides; Zhaowen Normal Luo of St. Jude for help with graphic design. Work at the FMX (17-ID-2) beamline is supported by the National Institute of Health, National Institute of General Medical Sciences (P41GM111244), and by the DOE Office of Biological and Environmental Research (KP1605010), and the National Synchrotron Light Source II at Brookhaven National Laboratory is supported by the DOE Office of Basic Energy Sciences under contract number DE-SC0012704 (KC0401040). Data were also collected at Southeast Regional Collaborative Access Team (SER-CAT) 22-ID (or 22-BM) beamline at the Advanced Photon Source, Argonne National Laboratory. SER-CAT is supported by its member institutions (see www.ser-cat.org/members.html), and equipment grants (S10_RR25528 and S10_RR028976) from the National Institutes of Health. This study was supported by ALSAC and NIGMS (R01GM129470) and St. Jude Cancer Center NCI (P30CA021765) funding to TM.

## Author contributions

G.S., C.D.G. and T.M. designed, performed, and analyzed experiments; G.S., S.J., A.A., and T.M. performed structural studies. C.R.G., D.E.M., S.V. and M.B.W. designed and performed experiments and analyzed and interpreted data; G.S. and T.M. wrote the paper. T.M. designed the project and interpreted data; all authors commented on the manuscript.

## Competing interests

The authors declare no competing interests.
