## [Peer Review File · Nature Communications]

Structural Basis of BAK Activation in Mitochondrial Apoptosis InitiationEditorial Note: Parts of this Peer Review File have been redacted.

Reviewers' Comments:

Reviewer #1:

Remarks to the Author:

Singh et al present a number of experiments addressing the question of BAK activation and aimed at teasing out the separate contributions of auto (BAK BH3) activation and direct (BID BH3) activation in apoptosis. Some interesting new data is reported: a structure of BAK BH3 bound to BAK, a BAK mutagenesis analysis that shows differential effects on auto- and direct-activation on liposomes and an analysis of BID mutants that show either enhanced activation or inhibition of activation. Experiments in BCL2 allKO cell lines reconstituted with BAK mutants provide some confirmation that under these extreme artificial conditions both modes of activation operate additively.

A more balanced introduction would provide a better setting for what follows. Only in the discussion do finally we read about prior studies that specifically and mechanistically link destabilisation of helix 1 to the activation of BAK and BAX, either with antibodies or with modified BIM BH3 peptides.

Figure 1.

I couldn't find any information to allow me to calculate the degree to which the DGS-NTA lipid was loaded by the various concentrations of BAK being added to the liposomes. The response in Fig1a is more like a step function than a titration. Could it be that excess BAK in solution is responsible for these observations?

The crystal structure of a BAK BH3 peptide bound to BAK (6uk9) was achieved by engineering a disulphide linkage between the peptide C-terminus and the protein. The authors observe that when the BH3 peptide binds to BAK some hydrogen bonds between helix 1 and the remainder of the protein are lost (including R42-D90 and E46-R137). The authors should note that R42 is L40 in murine BAK. A comparison with previous BH3 peptide complexes with BAK seeks to justify the novelty and relevance of these hydrogen bonds.

The critical new structure, 6uk9 – the BAK BH3 complex with BAK, is at modest resolution, precluding any analysis of solvent molecules. It's even debatable whether the claimed resolution of 3.1Å is justified here because the data is seriously incomplete (what are the R-factors in the outer shell?). The BIM complexes with BAK are higher resolution structures with various bound solvent molecules (one poorly ordered protein molecule in a crystal does not invalidate the findings from well-ordered ones). Although non-physiological, might these ligands reflect instabilities in their absence?

Without a picture overlaying 6uk9 and 5vww it is unclear how the absence of the solvent molecules in the BAK BH3 complex are accommodated. Is there disordered solvent in that case, or maybe even a cavity? Such a central matter to the claim of providing the "missing link" in BAK activation should be addressed.

Figure 2

The mutational analysis here contains interesting data on the failure of some mutants to autoactivate whilst being capable of direct activation by BID BH3. It is hard to disentangle ligand aspects from receptor aspects. Experiments using the BAK BH3 WT as a direct activator of the BAK mutants (and vice versa) should be performed to bolster the conclusions.

Use of 200nM BAK for the autoactivation seems right, but why use that concentration also for the BID activation? 50nM would seem more appropriate so as not to contaminate the experiment with autoactivation. Then, in Figure 2c, why use only 50nM BAK for the autoactivation?

The conclusion from Figure 2c that dye release does not correlate with cross-linking is confounded by the depletion of the monomer bands for L78A and I81R without much indication of what happened to them. The claim that direct and auto-activation have been uncoupled in these experiments should address the likelihood that mutants impaired in autoactivation may fail at the triggering step, not the dimerization step.

Figure 3

In fig 3b, the autoactivation result for I81A is not consistent with the liposome result in figure 2b.

The conclusion from these data should recognise both the "clean genetic background" and the use of mutated and N-terminally tagged versions of BAK. Both raise issues about the relevance to 'normal' cellular apoptosis.

Figure 4

Mutations of the BID BH3 to larger hydrophobic residues identifies reduction-of-function variants at W(1), Y(2), and W(2), and enhanced activation at M(3), Y(4) and possibly W(4). Decreases in function correlate with decreases in affinity of the mutant for BAK in an FP assay, and conversely those with increased activity have increased affinity. The outlier is W(5), which shows no increased activity but is the most potent binder, perhaps already hinting that activation occurs in an affinity window, above and below which no activation happens.

On page 25, "binding to pockets 3 and 4 either promotes BAK activation or inhibition ". I did not spot which one of these is inhibitory with high significance?

Figure 5

Based on the single mutant data, the double mutant M(3) W(5) was studied on the basis that it might have yet further enhanced affinity for BAK. Indeed it is 30 fold tighter but goes against the trend of increased activity seen in the single mutants. A crystal structure of this mutant BID BH3 bound to BAK (6uk8) reports the same hydrogen bonds as seen in the BAK BH3 complex with BAK (6uk9), but at higher resolution. A figure overlaying these structures would have been very helpful, given the complete absence of solvent from 6uk9. For that matter, what about a structure of the wild-type BID BH3 complex? That is surely the best comparator.

The author's 'belief' that this is "the first trustworthy....." is of little interest without supporting analysis of the solvent structure associated with wild-type BID BH3 bound to BAK. What makes artificial mutants trustworthy where wild-type structures with artificial solvent molecules are not?

The W(3) W(5) double mutant was examined next and it has further enhanced affinity for BAK. The authors express surprise that this peptide now shows inhibitory properties, yet the trend might have been detected in the prior double mutant whose activity was similar to wild-type BID BH3. It is also evident in the study of the BIM peptides where potent inhibitory activity was reported with high affinity BIM variants. The structure of this mutant BID BH3 with BAK is said to have a large cavity, though no detail is provided. It should be. Is it free of solvent molecules? Where is it? How is the inhibitory mechanism different to the one described for BIM variants? Both seem to rely on anchoring helix 1.

Discussion

How might I81R abolish autoactivation but allow direct activation?

Extant data suggests an explanation.

I presume that in 6ku9, I81 on the peptide is directed towards R42 on the protein. I base this on the BIM structures where the Ile in the h3 position has been changed to engage R42.

The I81R mutation would be unfavourable for putting this mutated BH3 into BAK. However, in the core dimer structure of BAK, which some argue is essential for poration, helix one has departed, so the I81R is permissive in this context. The flaw in this model should be exposed if the authors still find it 'remarkable that mutations in the BAK BH3 and the activation groove completely abolish autoactivation (eg I81R) but not poration'.

With respect to data and discussion about BAK inhibitors, everything presented is with liposomes. The BIM BH3 study addressed also the problem of bypassing the pro-survival proteins. This work falls short of tackling this problem, which is central to progressing effective BAK inhibitors.

What is Iyer et al (2020).

Reviewer #2:

Remarks to the Author:

Singh et al. report several new structures of BAK and BID BH3 complexes with BAK, revealing potential structural distinctions that could underlie an autoactivation mechanism that involves destabilization of the alpha-1 helix. The authors then perform peptide and protein mutagenesis studies, involving in vitro liposomal and cellular systems in an effort to test the structural findings and determine functional/physiologic relevance. Whereas the structural studies are concrete and lead to interesting hypotheses and potential mechanistic insights, the biochemical and cellular mechanistic validation studies are challenged by a combination of inconsistent findings and subtle effects in experiments that have small dynamic ranges. The latter produces a tension between "statistical significance" and biological meaning, and does not allow for definitive conclusions ("Our comprehensive understanding of direct activation, autoactivation, and inactivation provides an update framework..."). Although this paper could be suitable for publication in Nature Communications, experimental revisions and key changes in scientific presentation would be required.

1) Distinguishing between direct and autoactivation: The authors rely on specific mutations to distinguish between the two processes, but unfortunately the mutations do not provide a clear picture. For example, some mutants do or don't seem to have autoactivation behavior, but the role of these mutations on direct activation is less clear. For example, I81A, I81R, and D83A don't appear to autoactivate but there is direct activation impairment as well – this is likely due to deficiency in both processes, making it hard to tease apart (Fig. 2b) and the assay in Fig. 2c doesn't help because it also doesn't detect or distinguish between the two processes. There is also inconsistency when comparing liposomal and thermal stability results (Fig. 2b and S2c). For example, I81A and D83A do not appear to auto-activate and are impaired in direct activation as well by liposomal assay, but counter-intuitively these mutants are the least stable by thermal stability assay. Conversely, I85A both autoactivates and directly activates, but counter-intuitively shows relatively increased thermal stability. This, in turn, complicates interpretation of the cellular work, which is meant to validate the in vitro findings. The clearest cut mutations of the group for comparison appear to be V74A (retains autoactivation and essentially a normal response to direct activation) and L78A (defective in autoactivation but preserves direct activation, and has comparatively greater thermal stability – a logical correlation). Unfortunately, a V74A vs. L78A comparison is not translated to cellular validation studies. Instead, other mutations are carried forward that have inconsistent correlations or less distinguishing behaviors. In order to build a case for the mechanistic hypotheses catalyzed by the structures, a rigorous pair or two of distinguishing (rather than ambiguous) mutants should be tested and compared in in vitro and cellular studies.

2) Drawing conclusions based on statistically significant results from experiments that show small

dynamic ranges: A major concern is whether conclusions are appropriately drawn based on in vitro and cellular data that have very narrow dynamic ranges. See, for example, the data and y-axes of Figures 1d, S1d, 3c-d, 4c, S4f, S5b. I am concerned that the reader (e.g. biologist, biostatistician) would view such distinctions as having little to no biological meaning. This concern is exacerbated by the double digit micromolar binding affinities of some of the interactions, and high doses required to elicit even these subtle changes in the assays (e.g. 50 micromolar dosing in Fig. 1d to see a difference between ~8 and 15 AUC units). A key control missing from the cellular data (e.g. Fig. 3) is the inclusion of responses to reconstitution with WT BAK. Together these concerns impact the conclusions drawn regarding the distinguishing features of the two modes of activation and their proposed cooperativity.

3) Density of experimentation and presentation: The authors have clearly done an enormous amount of work and should be commended. It will be important, however, to improve the clarity of presentation for the reader (there's a lot to work through). For example, the insufficiency of prior structures in examining alpha-1 destabilization is important for showcasing the novelty of the structural work, but the presentation was hard to decipher from Table S2 and Fig S2C. Other areas for revision include: improve labeling of crystal structures, reduce "fading" of structures with depth, increase size of structures whenever possible, avoid color on color depictions (e.g. orange on orange in Fig. 1e), include PDB codes in figures, add molecular weight markers to all blots, better define BAK6t/7t in text or figure legends, clarify why the AUC calculation changes between experiments (e.g. 0-88 vs. 24-88: this should be standardized throughout and preferably 0-90 min without data exclusion), improve labeling of liposomal release subpanels (include concentrations, distinguishing features between groups of subpanels). The authors should clarify the caveats that come with comparing structural changes observed in the context of disulfide linked BH3 peptides (could the disulfide linking itself be having a structural effect?) and when comparing changes between a structure with and without a disulfide linked BH3 peptide (e.g. the two different structures with linked and unlinked BID mutant peptides). The introduction or discussion would benefit from putting the current work into context with the related body of knowledge about BAX autoactivation. Other specific points for clarification:

Figure 1 – consider overlaying structures in 1F to better demonstrate the differences in stabilization; clarify the hydrogen bonds in the plots (there appears to be more hydrogen bonds than what is listed in red/black in the header); include RMSD difference between the structures in 1F; are there other structural differences in BAK aside from what is shown by the BH3-facing view?

Figure 2 – Figure 2b and 2c use different amounts of BAK; what are the implications of examining a 4-fold different dose for the intended comparison between 2b and 2c?

Figure S2 – In S2D, I think there is a typo in the right portion of the figure, where "low" B factor should instead say high B factor.

In summary, the structural work is valuable and the mechanistic insights potentially important. If the authors could focus on the cleanest mutations and functional results with the greatest phenotypic/biological consequences, the study could be strengthened and more convincing to the reader. This reviewer is supportive of the work and encourages the authors to tighten up the story.

Reviewer #3:

Remarks to the Author:

The authors have studied the role of autoactivation on Bak activity compared to direct activation. They characterized the binding, activation capacity and structure of Bak BH3 peptides binding to Bak in vitro and using liposome assays. Thanks to the high resolution of the structures in the helix 1 region

compared to previous Bak complexes with BH3 peptides, they identified the residues involved in the destabilization of helix 1 that leads to Bak activation. Based on these structures, they designed a number of Bak mutants aimed at disentangling autoactivation from direct activation both in vitro and in cells lacking other Bcl-2 proteins. Based on the structures, the authors also designed Bid BH3 peptides with improved binding to Bak, of which they determined the structure of one with highest binding affinity that confirmed the destabilization of helix 1 during Bak activation, and also suggested that stabilization of helix 1 would instead lead to inhibition. These findings are novel and of high interest for the field.

However, a number of issues need to be addressed to strengthen the conclusions:

1. Peptide tethering was necessary to obtain the high resolution structures. How can the authors discard that this chemical modification does not lead to artefactual conformations? Could it be that the lack of high resolution in the region of helix 1 in previous structures is related to high flexibility of the region and/or to poorly defined interactions?
2. The authors designed a series of Bak mutants in the BH3 region to dissect autoactivation from direct activation, and indeed they found different activities in these mutants. But how are these mutations affecting the formation of the core domain in active symmetric Bak dimers? Interfering with the structure of active Bak dimers would also lead to a loss in activity, as seen by the lower activity when activated by Bid BH3. How would it compare to the Bak BH3 peptide?
3. In the end the cooperativity for Bak autoactivation is much more pronounced than when activating with Bid BH3, which could lead to the differences in the activity in figure 2b. These experiments are all done at one or two concentrations. It could all be a matter of concentration ranges (for example by being in some experiments under saturating conditions). To control for this, the authors should titrate protein/peptide concentrations systematically to expand figure 3 and determine the change in IC50 by plotting % permeabilization vs concentration.
4. For clarity, it would be helpful if the authors plot and compare the structures of the asymmetric and symmetric interactions between Bak molecules that they are proposing, highlighting the different interactions involved in each case.
5. In Figure 3b, how come I81A and D83A still have activity? It all seems a matter of kinetics rather than killing ability. In Figure 3b, why does QVD have so little inhibitory activity for V74A? How come QVD can't block cell death under autoactivation conditions? Is it possible that in I81R the core domain is also affected? In Figure S4 e, the bands for Bid cleavage are very faint and a negative control is missing, so that one cannot conclude that there is Bid activation.
6. For Figure 4, how do the authors explain the lack of correlation between Bid BH3 binding affinity of the different mutants to Bak and direct activator activity? In 4c it would be very helpful again to titrate several peptide concentrations and create graphs as in 4d. Why is the permeabilization of wt so low?
7. Several parts of the manuscript are described in a rather complex way and the text could be improved for clarity.

We thank the reviewers for genuine interests and heroic efforts in evaluating our study, criticisms, and great suggestions. Given key overlapping requests by the three reviewers we summarize the main points addressed in our rebuttal:

1. In response to reviewer #1, we further validated asymmetric "BH₃-in-groove" autoactivation by comparing i) mutant BAK BH₃ ligands+WT BAK receptor; and ii) WT BID and WT BAK BH₃ ligands+mutant BAK receptors. For this analysis we performed dose response curves, as requested by reviewer #3, which have revealed new insights into the extent of activation vs membrane permeabilization. These data confirm our original observations and interpretations with full-length protein. Point ii) revealed a significant difference in activation efficiency by WT BID BH₃ and WT BAK BH₃, with the latter peptide being ~one order of magnitude more potent despite similar binding affinities for BAK. These results also suggest that activation of a small fraction of BAK is sufficient to permeabilize membranes.
2. We performed additional and consolidated existing comparative analyses of structures of BH₃-bound BAK complexes, including cavity searches in these complexes. Our conclusion is that activation and inactivation by BH₃ ligands correlates well with destabilization and re-stabilization of helix 1 contacts at the bottom of the groove, respectively, but not with cavity induction at the groove.
3. Our attempt to investigate apoptosis by WT or L78A BAK in BCL2allKO HCT116 cell-based system, as suggested by reviewer #2, was unsuccessful because of lack of expression and degradation, respectively. However, we performed new normalization of cell death data to account for the BAK protein level and we noticed a striking parallel between liposome permeabilization and apoptotic response for several mutants that expressed successfully including V74A, I81A, and D83A.
4. We revised presentation of the thermal shift data, which was too elusive and confused reviewers.
5. We present crosslinking analysis at 200 nM protein as suggested by reviewers #1 and #2.
6. We added structural comparison of the BAK BH₃-BAK complex (asymmetric BH₃-in-groove dimer) with helix2-5 core dimer (symmetric BH₃-in-groove dimer) although we have no way of testing the effect of mutations on the core dimer functionally as these helices alone are not active. As reviewer #3 noted, the core dimer is expected to be impacted by mutagenesis in the BH₃ and activation groove.
7. We revised the presentation (changes tracked) as suggested by all reviewers.
8. Unfortunately, despite numerous attempts we did not obtain crystals for WT BID BH₃-BAK complex as suggested by reviewer #1.

By addressing all comments raised by reviewers we considerably improved our manuscript. We hope that reviewers find our revised version acceptable for publication. Below we included a detailed point-by-point response.

Reviewer #1 (Remarks to the Author):

Singh et al present a number of experiments addressing the question of BAK activation and

aimed at teasing out the separate contributions of auto (BAK BH₃) activation and direct (BID BH₃) activation in apoptosis. Some interesting new data is reported: a structure of BAK BH₃ bound to BAK, a BAK mutagenesis analysis that shows differential effects on auto- and direct-activation on liposomes and an analysis of BID mutants that show either enhanced activation or inhibition of activation. Experiments in BCL2 allKO cell lines reconstituted with BAK mutants provide some confirmation that under these extreme artificial conditions both modes of activation operate additively.

We thank this reviewer for the assessment and in-depth review of our manuscript.

A more balanced introduction would provide a better setting for what follows. Only in the discussion do finally we read about prior studies that specifically and mechanistically link destabilisation of helix 1 to the activation of BAK and BAX, either with antibodies or with modified BIM BH₃ peptides.

We agree with this reviewer. In the introduction we now cover these aspects detailing prior knowledge for the involvement of helix 1 in the activation of BAK and BAX.

Figure 1.

I couldn't find any information to allow me to calculate the degree to which the DGS-NTA lipid was loaded by the various concentrations of BAK being added to the liposomes. The response in Fig1a is more like a step function than a titration. Could it be that excess BAK in solution is responsible for these observations?

The reviewer is absolutely right: we did not explicitly specify the DGS-NTA(Ni) concentration. We have corrected this oversight now. The liposome permeabilization assay contains 8% DGS-NTA(Ni) (molar ratio), 4% cardiolipin, 45% phosphatidylcholine, 25% phosphatidylethanolamine, 9% phosphatidylinositol, 9% phosphatidylethanolamine, as stated in our methods section. The **final concentration of DGS-NTA(Ni) in our assays is 2.5 μM**, which is higher than tested BAK concentrations (0.05 μM, 0.1 μM, 0.2 μM, 0.4 μM, 0.8 μM) even when considering equal partition of this lipid in the inner and outer leaflet of the liposome bilayer [i.e. 1.25 μM DGS-NTA(Ni) lipid in each leaflet of the bilayer]. We added the final concentration of DGS-NTA(Ni) in the relevant methods and results section related to Figure 1a. Therefore, we reason that excess BAK in solution is most likely not responsible for these observations.

The crystal structure of a BAK BH₃ peptide bound to BAK (6uk9) was achieved by engineering a disulphide linkage between the peptide C-terminus and the protein. The authors observe that when the BH₃ peptide binds to BAK some hydrogen bonds between helix 1 and the remainder of the protein are lost (including R42-D90 and E46-R137). The authors should note that R42 is L40 in murine BAK.

This is a good point we addressed by describing this difference between human and murine BAK in Figure 6d, which shows alignment of BAK sequences from different species. It turns out that rodents including mouse and rat have an L instead of R₄₂, which is otherwise widely conserved. The buried network must rearrange differently in rodents compared to other species upon direct activation. This unfortunate difference makes one wonder how accurate mouse models are at probing the details of human pathophysiology.

A comparison with previous BH₃ peptide complexes with BAK seeks to justify the novelty and relevance of these hydrogen bonds.

The critical new structure, 6ukg – the BAK BH₃ complex with BAK, is at modest resolution, precluding any analysis of solvent molecules. It's even debatable whether the claimed resolution of 3.1Å is justified here because the data is seriously incomplete (what are the R-factors in the outer shell?). The BIM complexes with BAK are higher resolution structures with various bound solvent molecules (one poorly ordered protein molecule in a crystal does not invalidate the findings from well-ordered ones). Although non-physiological, might these ligands reflect instabilities in their absence?

We thank the reviewer for detailed observations and thoughtful questions.

- As the reviewer noted, the resolution of BAK BH₃-BAK complex is modest precluding any analysis of solvent molecules. We have now added the R-factors in the outer shell in Table 1.
- Additionally, we modified this section of our results to be inclusive of previous work with BIM BH₃:BAK complexes, as the reviewer is absolutely right that the presence of ligands may reflect instability of BAK in complex with activating BIM BH₃ peptides. As suggested by all reviewers, we added a comparison among BH₃:BAK complexes in new Figures 6, S8, S9. Based on this analysis we conclude that all complexes point to a unifying view for the mechanism of BAK activation—that of BH₃ activator induced destabilization of helix 1 contacts at the bottom of the groove. This question was also raised by reviewer #3.

Without a picture overlaying 6ukg and 5vww it is unclear how the absence of the solvent molecules in the BAK BH₃ complex are accommodated. Is there disordered solvent in that case, or maybe even a cavity? Such a central matter to the claim of providing the “missing link” in BAK activation should be addressed.

We present an overlay of 6ukg with several BH₃-bound BAK complexes including 5vww in Figure 6. Additionally, we calculated cavities for crystal structures of all activated BH₃:BAK complexes in Figure S8. Our analysis indicates that while we observe cavities in activated BIM BH₃-BAK complexes (5vww and 5vww), we do not observe similarly large cavities in any of our complexes. Therefore, cavity formation is not a universal mechanism of BAK destabilization in direct activation. It is important to note that the structures of BIM BH₃:BAK complexes have been determined using domain swapped BAK dimers, which may introduce artifactual cavities

induced by swapping and crystal-mounted pressure on the swapped dimers (i.e. helices 5 and 6, which are the point of swap, have been observed in alternate conformations in different crystal forms). In contrast, all of our structures have been done on monomeric BAK complexes.

Figure 2

The mutational analysis here contains interesting data on the failure of some mutants to autoactivate whilst being capable of direct activation by BID BH₃. It is hard to disentangle ligand aspects from receptor aspects. Experiments using the BAK BH₃ WT as a direct activator of the BAK mutants (and vice versa) should be performed to bolster the conclusions.

These are great points and suggestions. By addressing them we uncovered a hierarchy of direct activation and autoactivation in BAK-mediated membrane poration.

- By testing activation of WT protein receptor by mutant peptide ligands, we discovered that M71A and V74A BAK BH₃ peptides, which have missense mutations that do not significantly affect autoactivation in full-length BAK (hence “WT-like”, Figure 2b), activate WT BAK receptor as readily as WT BAK BH₃ ligand (Figure 2e). We also tested missense mutations I81R and I85R in BAK BH₃ peptide ligand, which in full-length BAK are resistant to autoactivation even at the highest doses tested, to discover that they are fully inactive in activation of the WT BAK receptor.
- We selected the WT-like V74A, as well as autoactivation-impaired mutants I81A, I81R, and D83A to test mutant protein receptor activation by WT peptide ligands. These are the mutants that we also interrogate in cells in Figures 3, 7, S5 [Note that L78A, which would have been an ideal mutant to test in cells, as suggested by reviewer #2, is degraded in cells (see immunoblot *vide infra*). Thus, we excluded it from *in vitro* analysis]. At 50 nM V74A BAK behaved very similar to WT BAK when activated by WT BID BH₃ or WT BAK BH₃ peptide ligands. Remarkably, WT BAK BH₃ peptide is more potent than WT BID BH₃ at lower peptide levels as suggested by EC₅₀ analysis of liposome permeabilization (EC₅₀ 50-200 nM vs 1500 nM, respectively). The EC₅₀ for BAK BH₃ is several orders of magnitude lower (~50-200 nM) than the binding affinity constant of this peptide for BAK (K_D~70 μM), whereas the EC₅₀ for BID BH₃ (1.5 μM) is much closer to its observed binding affinity constant for BAK (K_D~20-60 μM). These observations suggest that a very low level of activation is sufficient to activate BAK in membrane permeabilization, and that once BID activates BAK (**direct activation**), active BAK is more efficient in activation of dormant BAK *in trans* (**autoactivation**); this reveals a direct activation → autoactivation hierarchy in BAK activation. In contrast, at 50 nM protein receptor neither WT BID BH₃ or WT BAK BH₃ ligand could promote liposome permeabilization by the autoactivation impaired mutants. Related to the next point raised by this reviewer, we tested at 200 nM D83A, which was similarly activated by WT BID BH₃ or WT BAK BH₃ ligands. This mutant exhibited EC₅₀ ~5-7 μM in liposome permeabilization when activated by the WT BH₃ ligands, although it does not directly participate in BH₃ ligand binding (D83 is located on the periphery of the

activation groove Figure 2a), suggesting that this mutant is likely impaired in asymmetric BH₃-in-groove binding from the ligand side.

Thanks to these suggestions, we have validated the significant contribution of BAK autoactivation to membrane permeabilization and revealed a hierarchy in BAK activation.

Use of 200nM BAK for the autoactivation seems right, but why use that concentration also for the BID activation? 50nM would seem more appropriate so as not to contaminate the experiment with autoactivation.

As addressed above, none of the autoactivation impaired BAK mutants tested are active at 50 nM in the presence of WT BID or BAK BH₃ peptide. We therefore used 200 nM protein receptor to test i) direct activation and *in trans* activation of D83A BAK by WT BID BH₃ and WT BAK BH₃ peptide ligands, respectively, in Figures 2f, S4d, S4e; ii) direct activation of I81R BAK by M(3)W(5) BID BH₃ peptide ligand in Figures 7c, S10d, S10e.

Then, in Figure 2c, why use only 50nM BAK for the autoactivation?

We present crosslinking at 200nM BAK for WT, WT-like V74A, and autoactivation impaired I81A, I81R, and D83A in Figure S3e.

The conclusion from Figure 2c that dye release does not correlate with cross-linking is confounded by the depletion of the monomer bands for L78A and I81R without much indication of what happened to them.

We present dye release and crosslinking with WT and BAK mutants at 200 nM as suggested by this reviewer in the previous point. We observe less crosslinking for the autoactivation impaired mutants I81A, I81R, and D83A compared to the WT and WT-like V74A mutant in the absence and presence of BID. Regarding the different levels, we see that each treatment condition exhibits a different level of monomeric BAK despite the loading being the same. We explain the depletion in monomeric BAK with [+ GA + LUV - BID] treatment as being caused by alteration in the accessibility to Ab-1 antibody epitope at the N-terminus of BAK: the N-terminus NH₃ is crosslinked to the only lysine NH₃ group in BAK (K113) inducing a strain in the epitope that affects Ab-1 binding. We observe similar depletion in the monomer band when using Ab-1 to detect oxidized WT BAK with the C14-C166 internal disulfide bond (see Figure 4 in Singh and Moldoveanu Methods Mol Biol. 2019;1877:185-200). We can speculate that the [-GA -LUV -BID] treatment induced a much tighter monomer band because lipids from LUVs bind BAK in the [-GA +LUV -BID] resulting in aberrant broader migration of BAK monomer on SDS-PAGE. This systematic depletion is observed for all [-GA -LUV -BID] samples.

Therefore, our revised interpretation is that the inability of BAK to form asymmetric dimers in the autoactivation impaired mutants precludes crosslinking observed with the WT and the WT-like V74A mutant, even though these mutants are partially activated by BID BH₃.

The claim that direct and auto-activation have been uncoupled in these experiments should address the likelihood that mutants impaired in autoactivation may fail at the triggering step, not the dimerization step.

There are multiple aspects to this point that we addressed as follows, and which the reviewer also refers to below (e.g. first bullet point):

- In Figure S3d (original or new) we address the stability of mutants by thermal shift assays. This experiment is meant to test whether gross changes in melting temperature (T_m) introduced through specific mutations may affect the initial triggering of BAK in the absence of activation by BID BH₃ (i.e. autoactivation). The hypothesis is that lowering the T_m could make BAK more susceptible to initial triggering during autoactivation, while increasing the T_m could render autoactivation much more difficult thermodynamically possibly explaining why some mutants are resistant to autoactivation. We observed that I81A and D83A have significantly lower T_m compared to WT, suggesting that they may undergo “initial triggering” with “ease”, yet these mutants are severely impaired in autoactivation compared to WT. In contrast, the rest of the mutants exhibit higher T_m than WT and thus their stability could possibly hamper the initial triggering. However, the most inhibited mutant in autoactivation, I81R, exhibits a modest $<2^\circ\text{C}$ difference in T_m compared to WT. Taken together, this analysis suggests that the underlying mechanism for the observed autoactivation or lack thereof does not correlate well with mutant thermal stability, while correlating quite well with the BH₃-in-groove asymmetric dimerization. What is “initial triggering”? We do not know. It may be BAK spontaneously unfolding at high doses, which would reveal the active BAK BH₃ that can propagate autoactivation *in trans* through asymmetric BH₃-in-groove dimerization. It may also be that lipids promote BAK unfolding at high protein doses. We briefly speculate this last aspect in the discussion as suggested by a recent study (ref. 35).
- If the reviewer refers to the triggering step as the activation by BID BH₃, we tested D83A BAK receptor (Figure 2f), which does not directly participate in binding the BH₃ peptide in complexes of BAK with BAK BH₃ or BID-like BH₃ peptide ligands; we also tested I81R BAK receptor (Figure 7c), which compared to WT BAK exhibits marginally decreased affinity for activating M(3)W(5) BID BH₃ peptide (0.70 μM vs 1.34 μM). Remarkably, both of these mutants exhibit $\sim 5 \mu\text{M}$ EC₅₀ in liposome permeabilization upon direct activation suggesting that the asymmetric dimerization is severely impaired.
- It is conceivable that autoactivation-impaired mutants also affect the putative BH₃-in-groove symmetric dimerization, which has been proposed to be essential in mitochondrial poration. Our study does not address this point directly, but we provide assessment for the possible effect of mutations for this state in Figure S3c. Without further experimentation we conclude that the BH₃-in-groove symmetric dimer must be refractory to mutations that fully impair autoactivation to allow membrane poration as observed upon direct activation. We note that the $\alpha 2$ - $\alpha 5$ core dimer is inactive functionally precluding testing of mutations in the context of purified protein. In the absence of structural analysis of full-length active

BAK species associated with membranes we cannot further reconcile these results, which will be addressed in future studies.

Figure 3

In fig 3b, the autoactivation result for I81A is not consistent with the liposome result in figure 2b.

To address this very good point, we further analyzed the cell death data by including normalization for the level of BAK protein, essentially dividing the SYTOX Green counts/confluence (Figure 3c) by the mCherry counts (which represent mCherry-BAK fusion levels, Figure 3d). The normalized cell death results (Figure 3d 100 ng/mL Dox) are very consistent with liposome permeabilization at 800 nM presented in Figure 2d. Both I81A and D83A mutants exhibit some autoactivation at 800 nM in liposome permeabilization, and because they express at high levels compared to V74A in cells they induce significant cell death. Our normalization for BAK level resolved this issue.

The conclusion from these data should recognise both the “clean genetic background” and the use of mutated and N-terminally tagged versions of BAK. Both raise issues about the relevance to ‘normal’ cellular apoptosis.

These are great, relevant points that we now mention in the results section.

Although the Luo lab succeeded in expressing GFP-tagged WT BAK, our attempts at expressing mCherry-WT BAK failed in several attempts. This could be because this fusion is hyperactive, speaking of tagging the N-terminus. It is possible that the WT-like mCherry-V74A-BAK fusion expressed because of its partial degradation (see immunoblot in Figure S5a, and the blot included to address comments raised by reviewer #2, *vide infra*). The minimalist cell-based system has the additional caveat that BAK is under Dox control. To our knowledge, under normal conditions of cellular apoptosis gradual increase in the levels of BAK have not been reported as a significant contributor to mitochondrial poration.

Figure 4

Mutations of the BID BH₃ to larger hydrophobic residues identifies reduction-of-function variants at W(1), Y(2), and W(2), and enhanced activation at M(3), Y(4) and possibly W(4). Decreases in function correlate with decreases in affinity of the mutant for BAK in an FP assay, and conversely those with increased activity have increased affinity. The outlier is W(5), which shows no increased activity but is the most potent binder, perhaps already hinting that activation occurs in an affinity window, above and below which no activation happens.

The assessment is correct; however, before finalizing conclusions, the reviewer should consider our data presented in Figure 5, which describes the double substituted peptides that have

completely opposite functional outcomes despite similar affinities. It is the structural details that explain opposing functions.

On page 25, "binding to pockets 3 and 4 either promotes BAK activation or inhibition ". I did not spot which one of these is inhibitory with high significance?

In response to comments and suggestions by reviewer #3 we generated EC₅₀ analysis for the single mutant data and present it in Figure S6c. This analysis, which presents area under the curve for kinetic liposome permeabilization traces helps to illustrate that at most doses tested W(3) and F(4) achieve overall slightly less liposome permeabilization compared to WT, whereas M(3), Y(3), Y(4), and W(4) are more potent than the WT. The reviewer is correct in that high significance may only apply for Y(4) and W(4) (Figure 4c).

Figure 5

Based on the single mutant data, the double mutant M(3) W(5) was studied on the basis that it might have yet further enhanced affinity for BAK. Indeed it is 30 fold tighter but goes against the trend of increased activity seen in the single mutants. A crystal structure of this mutant BID BH₃ bound to BAK (6uk8) reports the same hydrogen bonds as seen in the BAK BH₃ complex with BAK (6uk9), but at higher resolution.

A figure overlaying these structures would have been very helpful, given the complete absence of solvent from 6uk9. For that matter, what about a structure of the wild-type BID BH₃ complex? That is surely the best comparator.

We overlaid the two complexes in Figure 6. The complex of BAK bound to WT BID has been on our wish list for as long as we have researched this field. Unfortunately, we could not deliver this structure yet, but we agree it would be the best comparator.

The author's 'belief' that this is "the first trustworthy....." is of little interest without supporting analysis of the solvent structure associated with wild-type BID BH₃ bound to BAK. What makes artificial mutants trustworthy where wild-type structures with artificial solvent molecules are not?

We have toned down the statement "the first trustworthy" and overall provide a much more thorough and inclusive account of all relevant structures in the field.

The point of our analysis is to design BH₃ peptides, which can be investigated structurally and either activate or inactivate BAK to reveal BH₃ ligand-induced BAK destabilization and re-stabilization, respectively. We succeeded in Figure 5.

The W(3) W(5) double mutant was examined next and it has further enhanced affinity for BAK. The authors express surprise that this peptide now shows inhibitory properties, yet the trend might have been detected in the prior double mutant whose activity was similar to wild-type

BID BH₃. It is also evident in the study of the BIM peptides where potent inhibitory activity was reported with high affinity BIM variants. The structure of this mutant BID BH₃ with BAK is said to have a large cavity, though no detail is provided. It should be. Is it free of solvent molecules? Where is it? How is the inhibitory mechanism different to the one described for BIM variants? Both seem to rely on anchoring helix 1.

There are multiple points and questions that the reviewer makes in the above paragraph, which we address point by point:

1. "The authors express surprise that this peptide now shows inhibitory properties, yet the trend might have been detected in the prior double mutant whose activity was similar to wild-type BID BH₃."

Indeed, this was an eureka moment for us when we examined high concentrations of M(3)W(5) BID BH₃ ligand (50 μM, >70× over the K_D ~0.7 μM, 98.6% bound) but do not observe inhibition. In contrast, 10 μM of W(3)W(5) BID BH₃ ligand (>33× over the K_D ~0.3 μM, 97.5% bound) was fully inhibitory. Thus, activation/inactivation are not strictly correlated with the affinity of the BH₃ ligands for BAK, although affinity probably helps inactivators more; most importantly activation and inactivation are about peptide binding induced conformational changes at the bottom of the activation groove to destabilize or stabilize helix 1, respectively.

2. It is also evident in the study of the BIM peptides where potent inhibitory activity was reported with high affinity BIM variants.

Table S1. Related to Figure 2, 4, 5, S3 and S6: Surface plasmon resonance data summary

		Bim-h3Pc	Bim-h3Glg	Bim-h3Glt	Bim-h0h3Glt
Bak	K _D	1.3 μM (± 0.5)	21 μM (± 3)	1.0 μM (± 0.1)	14.9 nM (± 0.3)
	ka (1/Ms)	8 (± 4) ×10 ³	2.8 (± 0.8) ×10 ³	8.5 (± 0.4) ×10 ³	4.5 (± 0.2) ×10 ⁴
	kd (1/s)	9.2 (± 0.2) ×10 ⁻³	6 (± 1) ×10 ⁻²	9 (± 1) ×10 ⁻³	0.70 (± 0.02) ×10 ⁻³
	Rmax	542 (± 66)	77 (± 37)	31 (± 14)	49 (± 11)
	Chi ² (RU ²)	2.60	0.16	0.21	0.32

To address this point, Table S1 from (Brouer *et al.* Mol. Cell 2017) summarizes BAK binding by the modified inhibitory BIM peptide, three of which are of modest affinity (1.0-21 μM) yet they inhibit BAK with excellent efficiency because they all form molecular glue complexes; in BIM BH₃-inhibitory complexes from the 2017 study, helix 1 is held in place through a double salt bridge to the unnatural amino acids Pc, Glg, or Glt at position (3) of BIM BH₃. Our W(3)W(5) BID BH₃ peptide achieves BAK inactivation through mimicking the apo BAK state within the electrostatic network at the bottom of the groove; thus, in our study BAK inhibition is achieved through a completely different molecular mechanism, overall preventing helix 1 destabilization despite inducing the greatest conformational changes observed in the activation groove of BAK. These distinctions have been made originally, and are emphasized better throughout the relevant results sections and discussion.

3. The structure of this mutant BID BH₃ with BAK is said to have a large cavity, though no detail is provided. It should be. Is it free of solvent molecules? Where is it?

In Figure S8 we present cavity analysis for all active BAK complexes determined by crystallography, as well as for apo BAK and W(3)W(5) BID BH₃:BAK complex. Our analysis predicts poor correlation between cavity formation and BAK activation, although we observe large cavities in the BIM BH₃ complexes as reported originally. We note that these complexes have been done using domain swapped dimers, which may introduce artifactual cavities compared to monomeric complexes as crystallization may exert pressure on the swap points. Structures for all of our complexes have been solved in the context of monomeric BAK.

4. How is the inhibitory mechanism different to the one described for BIM variants? Both seem to rely on anchoring helix 1.

Please see point 2 (*vide supra*).

Discussion

How might I81R abolish autoactivation but allow direct activation?

Extant data suggests an explanation.

I presume that in 6ku9, I81 on the peptide is directed towards R42 on the protein. I base this on the BIM structures where the Ile in the h3 position has been changed to engage R42. The I81R mutation would be unfavourable for putting this mutated BH₃ into BAK. However, in the core dimer structure of BAK, which some argue is essential for poration, helix one has departed, so the I81R is permissive in this context. The flaw in this model should be exposed if the authors still find it 'remarkable that mutations in the BAK BH₃ and the activation groove completely abolish autoactivation (eg I81R) but not poration'.

This is a great point, also raised by reviewer #3. We include comparative analyses with symmetric core dimers and interpret the possible effect of mutations rationally in Figures S3b, S3c. We feel that by adding this analysis we do not need to explicitly say that the core model is flawed; future studies will further probe this model.

Regarding the reviewer's point on how I81R may be accommodated within the groove of monomeric BAK or the core dimers, we want to remind the reviewer that I81R and I85R BAK BH₃ peptides do not activate WT BAK (Figure 2e). These mutants do not bind BAK by 2D-NMR (not shown in the paper but shown here); at this point we do not know how

they are accommodated in the core dimer structure, which exhibits a very similar groove structure as that of the BAK BH₃-BAK complex.

With respect to data and discussion about BAK inhibitors, everything presented is with liposomes. The BIM BH₃ study addressed also the problem of bypassing the pro-survival proteins. This work falls short of tackling this problem, which is central to progressing effective BAK inhibitors.

Our study does not aim to progress BH₃ peptides as BAK activators or inhibitors, and thus we do not think this point warrants further discussion in our manuscript. We merely use the peptides as proof of concept to understand what needs to happen at the level of BAK conformational changes to activate BAK and also to possibly inactivate BAK. As an aside and to promote full transparency in light of the study by Brouer et al (Mol. Cell 2017), the BIM BH₃ inhibitors of BAK designed in that study are activators of BAX, somewhat defeating the purpose of engineering selectivity. Nonetheless, we agree with this reviewer that the study by Brouer and colleagues beautifully illustrated what one may need to worry about when designing BAK-selective BH₃ peptide inhibitors.

What is Iyer et al (2020).

We updated this reference (ref. 45).

We thank this reviewer for a careful assessment of our manuscript. By attempting to address all of the questions raised by this reviewer, whenever possible with additional analyses, we believe that our manuscript is much improved.

Reviewer #2 (Remarks to the Author):

Singh et al. report several new structures of BAK and BID BH₃ complexes with BAK, revealing potential structural distinctions that could underlie an autoactivation mechanism that involves destabilization of the alpha-1 helix. The authors then perform peptide and protein mutagenesis studies, involving in vitro liposomal and cellular systems in an effort to test the structural findings and determine functional/physiologic relevance. Whereas the structural studies are concrete and lead to interesting hypotheses and potential mechanistic insights, the biochemical and cellular mechanistic validation studies are challenged by a combination of inconsistent findings and subtle effects in experiments that have small dynamic ranges. The latter produces a tension between "statistical significance" and biological meaning, and does not allow for definitive conclusions ("Our comprehensive understanding of direct activation, autoactivation, and inactivation provides an update framework..."). Although this paper could be suitable for publication in Nature Communications, experimental revisions and key changes in scientific presentation would be required.

We thank this reviewer for the positive and candid assessment. We made every effort to address the points raised by this reviewer whenever possible undertaking additional analyses.

1) Distinguishing between direct and autoactivation: The authors rely on specific mutations to distinguish between the two processes, but unfortunately the mutations do not provide a clear picture. For example, some mutants do or don't seem to have autoactivation behavior, but the role of these mutations on direct activation is less clear. For example, I81A, I81R, and D83A don't appear to autoactivate but there is direct activation impairment as well – this is likely due to deficiency in both processes, making it hard to tease apart (Fig. 2b) and the assay in Fig. 2c doesn't help because it also doesn't detect or distinguish between the two processes.

To address the issues raised by this reviewer, which were also raised by reviewer #1, we performed additional analyses to rationalize effects of mutants on direct activation, autoactivation (asymmetric BH₃-in-groove dimerization), as well as α 2- α 5 core dimerization (symmetric BH₃-in-groove dimerization), which we summarize in Figure S3c. In particular, in addition to testing full-length protein mutants in autoactivation, we tested i) mutant BAK BH₃ ligand + WT BAK receptor, and ii) WT BID BH₃ and WT BAK BH₃ with mutant BAK receptor. Through this analysis we revealed that BAK BH₃ is surprisingly more efficient than BID BH₃ at activating BAK, and that very little BAK activation is required to promote significant membrane permeabilization.

To address activation of the specific mutants mentioned by this reviewer, we know from the structural analysis of the BAK BH₃-BAK complex and our BID-like BH₃-BAK complexes that I81A and I81R directly impact binding of BH₃ ligands that contact I81 residue at pocket (5). How much the I81R mutation impacts binding of W(3)W(5) BID is suggested from our analysis presented in Figures 7c, S10d, which indicates that the WT and I81R BAK bound this peptide with K_D of 0.70 and 1.34 μ M, respectively; so I81R lowers affinity dissociation constant by ~2-fold. We could not obtain K_D values for the binding of WT BID BH₃ or WT BAK BH₃ to I81R BAK by ITC or SPR, but if we apply the same 2-fold difference compared to WT, their affinities would be ~130 μ M. On the other hand, structural analysis of all BH₃-BAK complexes indicate that D83 is not participating in binding the BH₃ ligand from the receptor side (e.g. Figure 2a), and thus D83A mutation in the receptor should not impact direct activation by WT BID BH₃ or asymmetric autoactivation by WT BAK BH₃. Our structural analysis of BAK BH₃-BAK complex demonstrates that D83 from the BAK BH₃ ligand side binds to R127 of the BAK receptor. Therefore, the D83A mutant should impact autoactivation exclusively from the ligand side. This mutant should also impact symmetric BH₃-in-groove dimerization (see Figure S3c). We hope that our reasoning is acceptable to the reviewer.

There is also inconsistency when comparing liposomal and thermal stability results (Fig. 2b and S2c). For example, I81A and D83A do not appear to auto-activate and are impaired in direct activation as well by liposomal assay, but counter-intuitively these mutants are the least stable by thermal stability assay.

This is a confusing point we rewrote to explain better. We addressed this point for reviewer #1. D83A is not impaired in direct activation based on our previous response. The point we want to make is that thermal stability does not correlate with mutants being competent in autoactivation (low or WT-like T_m) or impaired in autoactivation (high T_m). The important conclusion from this control experiment is that the mechanisms of asymmetric BH₃-in-groove dimerization is the significant contributor to autoactivation and not differences in thermal stability among mutants.

This is our response to comments raised by reviewer #1 regarding thermal stability:

"In Figure S3d (original or new) we address the stability of mutants by thermal shift assays. This experiment is meant to test whether gross changes in melting temperature (T_m) introduced through specific mutations may affect the initial triggering of BAK in the absence of activation by BID BH₃ (i.e. autoactivation). The hypothesis is that lowering the T_m could make BAK more susceptible to initial triggering during autoactivation, while increasing the T_m could render autoactivation much more difficult thermodynamically possibly explaining why some mutants are resistant to autoactivation. We observed that I81A and D83A have significantly lower T_m compared to WT, suggesting that they may undergo "initial triggering" with "ease", yet these mutants are severely impaired in autoactivation compared to WT. In contrast, the rest of the mutants exhibit higher T_m than WT and thus their stability could possibly hamper the initial triggering. However, the most inhibited mutant in autoactivation, I81R, exhibits a modest $<2^\circ\text{C}$ difference in T_m compared to WT. Taken together, this analysis suggests that the underlying mechanism for the observed autoactivation or lack thereof does not correlate well with mutant thermal stability, while correlating quite well with the BH₃-in-groove asymmetric dimerization. What is "initial triggering"? We do not know. It may be BAK spontaneously unfolding at high doses, which would reveal the active BAK BH₃ that can propagate autoactivation in trans through asymmetric BH₃-in-groove dimerization. It may also be that lipids promote BAK unfolding at high protein doses. We briefly speculate this last aspect in the discussion as suggested by a recent study (ref. 35)."

Conversely, I85A both autoactivates and directly activates, but counter-intuitively shows relatively increased thermal stability.

According to our above explanation, we should not rely on our intuition when it comes to interpreting thermal stability. The probable reason that this mutant is active in autoactivation is that I85 from the BAK BH₃ ligand side is juxtaposed with I85 from the receptor side in the asymmetric BH₃-in-groove BAK BH₃-BAK structure, and thus I85A mutation does not impact 2 possible binding sites in autoactive complexes as would, for instance, I81A. Nonetheless, I85R is completely inactive in autoactivation even at the highest doses tested as two arginine residues clash within the asymmetric BH₃-in-groove structure of the BAK BH₃-BAK complex. Moreover, we show that I85R BAK BH₃ is inactive in WT BAK activation (Figure S3e) and does not bind WT BAK by NMR (see diagram for NMR data shared with reviewer #1) supporting our model.

This, in turn, complicates interpretation of the cellular work, which is meant to validate the *in vitro* findings. The clearest cut mutations of the group for comparison appear to be V74A (retains autoactivation and essentially a normal response to direct activation) and L78A (defective in autoactivation but preserves direct activation, and has comparatively greater thermal stability – a logical correlation). Unfortunately, a V74A vs. L78A comparison is not translated to cellular validation studies. Instead, other mutations are carried forward that have inconsistent correlations or less distinguishing behaviors. In order to build a case for the mechanistic hypotheses catalyzed by the structures, a rigorous pair or two of distinguishing (rather than ambiguous) mutants should be tested and compared in *in vitro* and cellular studies.

We completely agree with the suggestion made this reviewer. It would have been ideal to compare the V74A and L78A mutants. Unfortunately, the L78A mutant is fully degraded in BCL2allKO HCT116 cells.

Although detectably expressed, based on the mCherry accumulation, L78A is inactive in apoptosis. We provide immunoblots of L78A along with several other mutants stably expressed in BCL2allKO HCT116. Some of the mutants including L78A, I85R, and even V74A, exhibit visible proteolytic degradation most likely on the BAK polypeptide side of the fusion, given that the mCherry signals are detectable (immunoblot was performed after 48 hr incubation with Dox + 40 μ M qVD). This is the main reason we proceeded with analysis of the stable mutants V74A, I81A, I81R, and D83A in Figure 3. We also want to point out that our multiple attempts at expressing mCherry-WT BAK in BCL2allKO HCT116 cells were unsuccessful, as we have been unable to detect a signal by imaging mCherry or by immunoblotting.

To strengthen our case for a comparison between liposome permeabilization and apoptosis, we performed another normalization by dividing SYTOX Green counts/confluence by the mCherry counts, which reflect the levels of the various m-Cherry-BAK mutant proteins. This data is now presented in Figures 3c, 3d, showing very close correlation between apoptosis and liposome permeabilization for I81A, I81R, and D83A (Figure 2d). We hope that this reviewer understands the limitations faced in expressing mCherry fusions of WT and mutant BAK in BCL2allKO HCT116 and will appreciate the striking parallels between *in vitro* liposome permeabilization and apoptotic response.

2) Drawing conclusions based on statistically significant results from experiments that show small dynamic ranges: A major concern is whether conclusions are appropriately drawn based

on in vitro and cellular data that have very narrow dynamic ranges. See, for example, the data and y-axes of Figures 1d, S1d, 3c-d, 4c, S4f, S5b. I am concerned that the reader (e.g. biologist, biostatistician) would view such distinctions as having little to no biological meaning. This concern is exacerbated by the double digit micromolar binding affinities of some of the interactions, and high doses required to elicit even these subtle changes in the assays (e.g. 50 micromolar dosing in Fig. 1d to see a difference between ~8 and 15 AUC units). A key control missing from the cellular data (e.g. Fig. 3) is the inclusion of responses to reconstitution with WT BAK. Together these concerns impact the conclusions drawn regarding the distinguishing features of the two modes of activation and their proposed cooperativity.

There are a number of points that the reviewer brings up in this paragraph:

- First, we would like to remind the reviewer that we could not express mCherry-WT-BAK in BCL2allKO HCT116 cells (see above immunoblot), but that we showed how mCherry-V74A-BAK behaves similar to WT in every assay tested. Thus, V74A results reflect what would happen with WT BAK in cells.
- Regarding statistical and biological significance, we believe that our additional analyses to calculate EC₅₀ values for liposome permeabilization and to normalize apoptosis to mCherry-BAK protein levels should help the reader better interpret and appreciate the biological significance of our results. A major conclusion is that the WT-like V74A is much more active than all mutants impaired in autoactivation in vitro and in cells.
- The reviewer is also specifically concerned about the affinity of BAK BH₃ for BAK (K_D ~67 μM) and biological relevance, yet this affinity is very similar to that of BID BH₃ for BAK (K_D ~62 μM) according to our SPR analysis. We do not think this is out of line with the vast literature indicating that weak affinity BH₃ ligands exert potent BAK activation. Moreover, we now provide EC₅₀ analysis for liposome permeabilization assays, which has revealed differences between BID BH₃ and BAK BH₃ ligands in activation of BAK receptor and has suggested possible hierarchy for BAK activation: once BID BH₃ activates a small fraction of total BAK, active BAK can significantly contribute to signal amplification through autoactivation by asymmetric BH₃-in-groove dimerization. This is observed in cells when direct activation cooperates with autoactivation (Figure 3e).

3) Density of experimentation and presentation: The authors have clearly done an enormous amount of work and should be commended. It will be important, however, to improve the clarity of presentation for the reader (there's a lot to work through).

We thank this reviewer for the kind note and the suggestions to make our presentation more accessible. We have worked on our presentation accordingly.

For example, the insufficiency of prior structures in examining alpha-1 destabilization is important for showcasing the novelty of the structural work, but the presentation was hard to decipher from Table S2 and Fig S2C.

We took the following steps to present this point better:

- As suggested by reviewer #1, we have included an introductory paragraph that details prior work implicating helix 1 dissociation in the activation of BAK and BAX.
- We have included a comparison between BAK BH₃-BAK structures and all other activated BAK structures as well as some of the inactive structures to highlight the known features and new discoveries related to alpha-1 destabilization and re-stabilization (Figures 6, S9).
- We agree that Table S2 is dense, but such tables always are, which is why we have it in the supplementary data; we keep it unchanged only for completion for anyone interested in the absolute values for distances of the electrostatic network involving helix 1. We believe that the visual depiction of the electrostatic networks for BH₃-BAK complexes is much easier to understand and compare in Figures 1, 5, and S9. We hope that the reviewer can better decipher the electrostatic networks in the new presentation. Movie S1 is also helpful.

Other areas for revision include:

- improve labeling of crystal structures,

We added the PDB to each structure and double checked that they are named consistently throughout

- reduce “fading” of structures with depth,

We have done this throughout

- increase size of structures whenever possible,

We have done this throughout

- avoid color on color depictions (e.g. orange on orange in Fig. 1e),

We modified colors throughout

- include PDB codes in figures,

The PDB codes are included

- add molecular weight markers to all blots,

Molecular weight markers have been added

- better define BAK6t/7t in text or figure legends,

These have now been better defined

- clarify why the AUC calculation changes between experiments (e.g. 0-88 vs. 24-88: this should be standardized throughout and preferably 0-90 min without data exclusion),

We present AUC 0-88 min (88 min) throughout. Related to the request not to exclude any data, the Simpson AUC algorithm uses multiples of 3 points for integration. Therefore, we typically exclude the 90 min data point (46th point) and calculate AUC for 0-88 min. In Figures 7b we present AUC data for shorter time intervals to quantify differences among mutants that exhibit changes in early kinetics.

- improve labeling of liposomal release subpanels (include concentrations, distinguishing features between groups of subpanels).

We marked the concentrations in the liposome permeabilization subpanels or the related legends.

- The authors should clarify the caveats that come with comparing structural changes observed in the context of disulfide linked BH₃ peptides (could the disulfide linking itself be having a structural effect?) and when comparing changes between a structure with and without a disulfide linked BH₃ peptide (e.g. the two different structures with linked and unlinked BID mutant peptides).

We reason that there are no caveats with the structures that are tethered through disulfide linkage. For the BAK BH₃-BAK complex (6ukg) this linker is not fully visible in any of the 10 complexes in the asymmetric unit suggesting that it is disordered (Figures S2c). Only in one of the two M(3)W(5) BID BH₃-BAK complexes in the asymmetric unit is the linker fully visible, again suggesting that the linker does not change the structure given that the overlay of the two complexes show very similar structures (Figure S7f). The differences between the structures of M(3)W(5) BID BH₃-BAK and W(3)W(5) BID BH₃-BAK complexes is mainly induced by their difference at position (3).

- The introduction or discussion would benefit from putting the current work into context with the related body of knowledge about BAX autoactivation.

We briefly discuss BAX as well as BOK autoactivation in the discussion. An excellent reference on the latest trends in BAX activation is now included (ref. 47).

Other specific points for clarification:

Figure 1 – consider overlaying structures in 1F to better demonstrate the differences in stabilization; clarify the hydrogen bonds in the plots (there appears to be more hydrogen bonds than what is listed in red/black in the header); include RMSD difference between the

structures in 1F; are there other structural differences in BAK aside from what is shown by the BH₃-facing view?

The structures were overlaid before they were split in Figure 1f and we are showing the key differences. Additionally, we show this overlay from two different views in Figure 6b. As seen in that figure there are no gross structural differences in BAK aside from those highlighted in Figure 1f. Our figure legends detail what the numbers stand for: they are the bonds that link helix 1 to the rest of the domain; these are now marked also with asterisks. The additional hydrogen bonds in the electrostatic network are included for completion to satisfy neutralization of charged residues in the core of the protein (as summarized in Table S2); they do not contribute to the bond number in the header.

Figure 2 – Figure 2b and 2c use different amounts of BAK; what are the implications of examining a 4-fold different dose for the intended comparison between 2b and 2c?

We present crosslinking analysis at 200 nM protein. We addressed this point for reviewer #1 as follows:

*“We performed new dye release and crosslinking with WT and BAK mutants at 200 nM as suggested by this reviewer in the previous point. We observe less crosslinking for the autoactivation impaired mutants I81A, I81R, and D83A compared to the WT and WT-like V74A mutant in the absence and presence of BID. Regarding the different levels, we see that each treatment condition exhibits a different level of monomeric BAK despite the loading being the same. We explain the depletion in monomeric BAK with [+ GA + LUV - BID] treatment as being caused by alteration in the accessibility to Ab-1 antibody epitope at the N-terminus of BAK: the N-terminus NH₃ is crosslinked to the only lysine NH₃ group in BAK (K113) inducing a strain in the epitope that affects Ab-1 binding. We observe similar depletion in the monomer band when using Ab-1 to detect oxidized WT BAK with the C14-C166 internal disulfide bond (see Figure 4 in Singh and Moldoveanu *Methods Mol Biol.* 2019;1877:185-200). We can speculate that the [-GA -LUV - BID] treatment induced a much tighter monomer band because lipids from LUVs bind BAK in the [-GA +LUV -BID] resulting in aberrant broader migration of BAK monomer on SDS-PAGE. This systematic depletion is observed for all [-GA -LUV -BID] samples.*

Therefore, our revised interpretation is that the inability of BAK to form asymmetric dimers in the autoactivation impaired mutants precludes crosslinking observed with the WT and the WT-like V74A mutant, even though these mutants are partially activated by BID BH₃.”

Figure S2 – In S2D, I think there is a typo in the right portion of the figure, where “low” B factor should instead say high B factor.

The reviewer is right. We corrected this error.

In summary, the structural work is valuable and the mechanistic insights potentially important. If the authors could focus on the cleanest mutations and functional results with the greatest

phenotypic/biological consequences, the study could be strengthened and more convincing to the reader. This reviewer is supportive of the work and encourages the authors to tighten up the story.

We greatly appreciate the criticisms raised by this reviewer. By addressing them all we considerably improved our study.

Reviewer #3 (Remarks to the Author):

The authors have studied the role of autoactivation on Bak activity compared to direct activation. They characterized the binding, activation capacity and structure of Bak BH₃ peptides binding to Bak in vitro and using liposome assays. Thanks to the high resolution of the structures in the helix 1 region compared to previous Bak complexes with BH₃ peptides, they identified the residues involved in the destabilization of helix 1 that leads to Bak activation. Based on these structures, they designed a number of Bak mutants aimed at disentangling autoactivation from direct activation both in vitro and in cells lacking other Bcl-2 proteins. Based on the structures, the authors also designed Bid BH₃ peptides with improved binding to Bak, of which they determined the structure of one with highest binding affinity that confirmed the destabilization of helix 1 during Bak activation, and also suggested that stabilization of helix 1 would instead lead to inhibition. These findings are novel and of high interest for the field.

We thank this reviewer for the kind summary.

However, a number of issues need to be addressed to strengthen the conclusions:

1. Peptide tethering was necessary to obtain the high-resolution structures. How can the authors discard that this chemical modification does not lead to artefactual conformations?

This is a great point we addressed also in response to a similar question raised by reviewer #2 as follows:

"We reason that there are no caveats with the structures that are tethered through disulfide linkage. For the BAK BH₃-BAK complex (6uk9) this linker is not fully visible in any of the 10 complexes in the asymmetric unit suggesting that it is disordered (Figures S2c). Only in one of the two M(3)W(5) BID BH₃-BAK complexes in the asymmetric unit is the linker fully visible, again suggesting that the linker does not change the structure given that the overlay of the two complexes show very similar structures (Figure S7f). The differences between the structures of M(3)W(5) BID BH₃-BAK and W(3)W(5) BID BH₃-BAK complexes is mainly induced by difference at position (3)."

Could it be that the lack of high resolution in the region of helix 1 in previous structures is related to high flexibility of the region and/or to poorly defined interactions?

The reviewer is right. Previous structures have hinted to poorly defined interactions involving helix 1 at the bottom of the activation groove. We now take a more unifying view of the destabilization of helix 1 interface observed in all determined structures.

Our response to a similar question raised by reviewer #1 is:

- *"Additionally, we modified this section of our results to be inclusive of previous work with BID SAHB:BAK and BIM BH₃:BAK complexes, as the reviewer is absolutely right that the presence of ligands may reflect instability in BAK in complex with activating BIM BH₃ peptides. Based on all three reviewers' comments, we added a comparison among all known BH₃ complexes with BAK in new Figures 6, S8, S9. We now conclude that all complexes point to a unifying view for the mechanism of BAK activation—that of BH₃ activator induced destabilization of helix 1 contacts at the bottom of the groove. This question was also raised by reviewer #3."*

2. The authors designed a series of Bak mutants in the BH₃ region to dissect autoactivation from direct activation, and indeed they found different activities in these mutants. But how are these mutations affecting the formation of the core domain in active symmetric Bak dimers? interfering with the structure of active Bak dimers would also lead to a loss in activity, as seen by the lower activity when activated by Bid BH₃. How would it compare to the Bak BH₃ peptide?

The reviewer is right that mutations that affect asymmetric BH₃-in-groove autoactivation should also impact symmetric BH₃-in-groove core dimerization. However, as long as enough direct activator is added, BAK is activated regardless of mutations (e.g. D83A and I81A) suggesting that these mutations are tolerated in BAK states downstream direct activation and autoactivation. The core dimer was expressed as a GFP fusion, which is inactive and cannot be tested functionally in that context. We therefore did not consider GFP- α 2- α 5 core dimer mutants experimentally. However, we have compared BAK BH₃-BAK complex with the core dimer in Figure S3b and show the predicted effect of mutations on the core dimer in Figure S3c. Understanding the role of mutation in the core dimer will be of great interest in future studies.

Regarding the comparison between BID BH₃ and BAK BH₃, we have now addressed this comparison as explained in the following point.

3. In the end the cooperativity for Bak autoactivation is much more pronounced than when activating with Bid BH₃, which could lead to the differences in the activity in figure 2b. These experiments are all done at one or two concentrations. It could all be a matter of concentration ranges (for example by being in some experiments under saturating conditions). To control for this, the authors should titrate protein/peptide concentrations systematically to expand figure S3 and determine the change in IC₅₀ by plotting % permeabilization vs concentration.

This is a great suggestion, similar to a suggestion raised by reviewer #1, which we addressed by performing dose response analysis for a select group of WT and mutant (V74A, I81A, I81R,

D83A) BAK protein receptors against WT BID BH₃ and WT and mutant (M71A, V74A, I81R, I85R) BAK BH₃ peptide ligands. We present EC₅₀ values for the liposome permeabilization assays (Figures 2e, 2f, S4).

4. For clarity, it would be helpful if the authors plot and compare the structures of the asymmetric and symmetric interactions between Bak molecules that they are proposing, highlighting the different interactions involved in each case.

We performed comparative analyses in Figure S3b, 3c and rationally interpreted the effect of mutations on each BAK state. We also explain more clearly how mutants impact the different states in the relevant results sections.

5. In Figure 3b, how come I81A and D83A still have activity? It all seems a matter of kinetics rather than killing ability.

This is a great question that reviewers #1 and #2 also raised. The simple answer is that the protein levels for the autoactivation-impaired mutants is quite high in BCL2allKO HCT116 cells. We addressed this issue as follows:

"To address this very good point, we further analyzed the cell death data by including normalization for the level of BAK protein, essentially dividing the SYTOX Green counts/confluence (Figure 3c) by the mCherry counts (which represent mCherry-BAK fusion levels, Figure 3d). The normalized cell death results (Figure 3d 100 ng/mL Dox) are very consistent with liposome permeabilization at 800 nM presented in Figure 2d. Both I81A and D83A mutants exhibit autoactivation at 800 nM in liposome permeabilization, and because they express at high levels compared to V74A in cells they induce significant cell death. Our normalization for BAK level resolved this issue."

Additionally, as reviewer #1 suggested, the mCherry tagging could further change the overall cellular activity of these mutants.

In Figure 3b, why does QVD have so little inhibitory activity for V74A?

These assays are rather long 48 hr. There is a possibility that cells are dying in a caspase-independent manner once the outer mitochondrial membrane has been compromised with V74A (qVD unable to efficiently block death). The slower death by the other mutants does not induce as much caspase-independent cell death. This is now proposed in the text and a reference for caspase-independent cell death downstream of mitochondrial poration is provided.

How come QVD can't block cell death under autoactivation conditions?

qVD blocks autoactivation as shown in Figure S5c. If the reviewer refers to the data in Figure 3e, S5f, we added qVD at 6 hr and we believe that there are cells that have already started

dying with caspases cleaving key apoptotic substrates. This is potentially why we observe trailing of cell death even in the presence of qVD. Additionally, as described in the above point, there may be caspase independent cell death downstream mitochondrial poration.

Is it possible that in I81R the core domain is also affected?

This is a good point. I81R most definitely should affect the core domain as indicated in our predictions for this mutant in Figure S3c. However, pushing this mutant to activation at higher doses of activating M(3)W(5) BID BH₃ achieves similar membrane permeabilization as WT BAK (Figure 7c).

In Figure S4 e, the bands for Bid cleavage are very faint and a negative control is missing, so that one cannot conclude that there is Bid activation.

We include BID immunoblots ± qVD in Figure S5e. qVD blocks BID processing by inhibiting caspase 8. We tried to perform the immunoblot at the end of the cell death assays but the signal is diminished beyond detection in the absence of qVD. This is the reason we performed immunoblotting at 90 min after TRAIL+CHX±qVD, when we detect proteolysis judged by tBID band. The blot at 3 hr after TRAIL+CHX±qVD treatment is included here but not in the paper.

6. For Figure 4, how do the authors explain the lack of correlation between Bid BH₃ binding affinity of the different mutants to Bak and direct activator activity? In 4c it would be very helpful again to titrate several peptide concentrations and create graphs as in 4d.

We present the EC₅₀ analysis for liposome permeabilization in Figure 4 and now summarize the activity and binding data in a table format (Figure 4e). We can explain the lack of correlation between activity and binding by considering that affinity does not equal activation or inactivation; instead, ligand-induced structural changes along with affinity can better define functional outcome. This is better demonstrated by the double substituted peptides and their complexes with BAK in Figure 5.

Why is the permeabilization of wt so low?

We do not think that WT is so low especially if one considers the data for AUC vs concentration in Figure S6c.

7. Several parts of the manuscript are described in a rather complex way and the text could be improved for clarity.

We thank this reviewer for deeply thoughtful comments. We hope to have tightened up the presentation and that we addressed all criticisms. In doing so we greatly improved the significance of our study.

Reviewers' Comments:

Reviewer #1:

Remarks to the Author:

Singh et al have made significant revisions to their manuscript, and referee-inspired additional experiments have bolstered their findings. The work is now publishable though I remain unconvinced of its suitability for Nat Comms. The language is considerably toned down, but are the findings truly novel or only incremental?

I apologise for introducing a new point that should have been raised in round 1. I think the authors should define their use of the terms direct activation (here meaning induced by a BID BH3 peptide) and autoactivation (here meaning induced by a BAK BH3 peptide, as an approximation to what may happen in real autoactivation when the BH3 is being presented in the context of full-length BAK). The distinction between real autoactivation and what is measured in these experiments should be highlighted.

The requested overlay of 6uk9 and 5vww is provided in figure 6, which is, however, completely uninformative. What is needed is an overlay with the right-hand panel of figure 1f, and similar pictures for the complexes with the two BID BH3 mutant complexes. Also, provide the structural comparison of the W3W5 BID BH3 mutant complex with apo BAK. I appreciate that when the pdb files are released all will be clearer, but the casual reader should be able to make sense of figure 6 (bc) in terms of the claims made for the novelty of 6uk9.

Disentangling the effects of BAK ligand mutants and BAK receptor mutants has produced interesting new findings.

In line 783 a novel mechanism of BAK inactivation is claimed. The answer to the question on the differences in this inhibitory mechanism to that described by others for BIM analogues is said to be in point 2 of page 9 of the rebuttal. But this only serves to make my point. The structural details may differ, but is the mechanism not the same - stabilising helix 1? The only known natural inhibitors (inactivators) of BAK are the antiapoptotic BCL-2 proteins, and a substantial body of data suggests they do this by sequestering the BAK BH3. The work with the BIM BH3 analogues showed a new mechanism, binding to the groove and stabilising helix 1. The present work provides just a second example of a structurally-characterised, non-natural inhibitory molecule. It is a stretch to claim too much for either one, though both suggest stabilisation of helix 1 to be important, and the idea that this could be achieved through somewhat different molecular interactions is neither surprising nor novel.

Table S2 has some errors, like H bonds between two H bond donors.
The Table in figure S3c is a very helpful summary of the findings.

Reviewer #2:

Remarks to the Author:

We appreciate the efforts and explanations provided by the authors in response to the reviews. Overall the structures are the highlight of the study and they suggest interesting mechanistic hypotheses. The weaknesses that remain are the biochemical and cellular studies that attempt to distinguish between the mechanisms of direct and auto activation. The high doses of ligands to induce effects, the subtlety of too many of the biochemical and cellular findings (very small differences with expanded y axes where statistical differences are not convincingly biologically significant), mutants that have mixed activities, and cellular studies limited by technical challenges (inability to compare with wild type Bak, mutants with cleanest differences in mechanism not evaluable in cells), leaves the reader insufficiently convinced by the data as presented. A focused paper on the interesting new structures and the

mechanistic insights they suggest seems preferable until the biochemical and cellular work can be shored up and experimentally clarified. Also, the paper is very dense and difficult to read. As more convincing data is compiled, the story may simplify and become more readable.

Reviewer #3:

Remarks to the Author:

The authors have addressed most of the key questions addressed by the reviewers and provided reasonable explanations why some issues cannot be experimentally tested. The manuscript remains very long and complicated to read, but it contains an impressive amount of work and a relevant take-home message that deserves publication in Nature Communications.

Point-by-point response

We thank the reviewers for evaluating our revised manuscript and for their constructive input. This revision preserves our original conclusions while improving analyses and presentation. We address their comments below.

REVIEWER 1

I apologise for introducing a new point that should have been raised in round 1. I think the authors should define their use of the terms direct activation (here meaning induced by a BID BH3 peptide) and autoactivation (here meaning induced by a BAK BH3 peptide, as an approximation to what may happen in real autoactivation when the BH3 is being presented in the context of full-length BAK). The distinction between real autoactivation and what is measured in these experiments should be highlighted.

We appreciate the suggestion and now distinguish BAK autoactivation measured in our studies as the asymmetric BH3-in-groove binding of BAK BH3 to the activation groove of BAK and have made this clear throughout the text and in our updated model (Figure 8).

The requested overlay of 6uk9 and 5vww is provided in figure 6, which is, however, completely uninformative. What is needed is an overlay with the right-hand panel of figure 1f, and similar pictures for the complexes with the two BID BH3 mutant complexes. Also, provide the structural comparison of the W3W5 BID BH3 mutant complex with apo BAK. I appreciate that when the pdb files are released all will be clearer, but the casual reader should be able to make sense of figure 6 (bc) in terms of the claims made for the novelty of 6uk9.

We have taken this request on board and have overlaid the new structures with apo BAK (pdb 2imt) in **Figures 1f, 6e, 6g** and have moved the original figures to **Figure S2d, S8g, S8i**. Additionally, we present overlays between BAK BH3 and M(3)W(5) BID BH3 activated BAK, M(3)W(5) BID BH3 and BIM-RT activated BAK, and W(3)W(5) BID BH3 and BIM-h3Pc-RT inactivated BAK in revised **Figure 7b**. These overlays better reveal conformational changes induced by BH3 ligands in the electrostatic network stabilizing helix 1 explaining BAK activation and inactivation.

In line 783 a novel mechanism of BAK inactivation is claimed. The answer to the question on the differences in this inhibitory mechanism to that described by others for BIM analogues is said to be in point 2 of page 9 of the rebuttal. But this only serves to make my point. The structural details may differ, but is the mechanism not the same - stabilising helix 1? The only known natural inhibitors (inactivators) of BAK are the antiapoptotic BCL-2 proteins, and a substantial body of data suggests they do this by sequestering the BAK BH3. The work with the BIM BH3 analogues showed a new mechanism, binding to the groove and stabilising helix 1. The present work provides just a second example of a structurally-characterised, non-natural inhibitory molecule. It is a stretch to claim too much for either one, though both suggest stabilisation of

helix 1 to be important, and the idea that this could be achieved through somewhat different molecular interactions is neither surprising nor novel.

We appreciate the reviewer's viewpoint but respectfully disagree. We clarify the distinction between W(3)W(5) BID BH3 and BIM-h3Pc-RT inactivated complexes by explaining that stabilization of helix 1 is through indirect contacts in the former (allosteric) and through direct contact in the latter "glue-like" complex. This distinction is now discussed in the highlighted region on pg. 16 being key for the design of small molecule BH3 mimetic.

Table S2 has some errors, like H bonds between two H bond donors.

The reviewer is correct in noting these errors, which reflect incorrect side chain rotamers found in the BID SAHB-BAK NMR structure (pdb 2m5b). We further note that this NMR structure exhibits heterogeneity in the electrostatic network distances among the 20 lowest energy models (a limitation of standard NOE-based NMR at capturing electrostatic contacts) and have therefore excluded this structure from **Table S2 and Figure S9**, which now focus exclusively on the crystal structures of BAK in complex with BH3 ligands. Removing pdb 2m5b from the analysis did not alter our conclusions.

REVIEWER 2

The weaknesses that remain are the biochemical and cellular studies that attempt to distinguish between the mechanisms of direct and auto activation. The high doses of ligands to induce effects, the subtlety of too many of the biochemical and cellular findings (very small differences with expanded y axes where statistical differences are not convincingly biologically significant), mutants that have mixed activities, and cellular studies limited by technical challenges (inability to compare with wild type Bak, mutants with cleanest differences in mechanism not evaluable in cells), leaves the reader insufficiently convinced by the data as presented.

We thank this reviewer for assessing our work and prompting us to provide additional insights on the system. We have invested significant effort and resources in designing new experimental strategies to address all the points raised. We believe that the biochemical and cellular findings have been strengthened, without altering our original conclusions. We provide answers to the specific comments below:

1. High doses of ligands to induce effects

The BH3 peptide ligands used to activate BAK in liposome permeabilization assays were at concentrations from 0.1 μ M for WT BAK BH3 to 1 μ M for WT BID BH3 (Figure 1d). This is consistent with our previous work [Moldoveanu et al NSMB (2013) 20:589], wherein BID BH3 is \sim 100x less potent than activated full-length BID (N/C BID). Similarly, lower activity for BH3 peptides compared to full-length proteins has been reported by others (see references 12-29 in

manuscript). We therefore believe that our observations are in line with known properties of BH3 peptide ligands.

Additionally, we show correlation of performance by BAK BH3 peptide ligands and full-length proteins in autoactivation. For instance, M71A and V74A mutations do not affect the activity of BAK BH3 peptide ligands which behave similar to WT BAK BH3 or full-length BAK (**Figure S3d**). In contrast, I81R and I85R mutant BAK BH3 ligands are inactive (**Figure S3d**).

2. Subtlety of too many of the biochemical and cellular findings (very small differences with expanded y axes where statistical differences are not convincingly biologically significant)

We thank the reviewer for this general criticism. We believe the reviewer refers to the data in original **Figure 3** in which we normalize cell death to the mCherry-BAK level obtained by IncuCyte imaging, resulting in very small differences in cell death between V74A, I81A, and D83A mutants visible with expanded y axis that are not convincingly biologically significant. We have now revised cell death analysis with mC-V74A-BAK and mC-D83A-BAK by correlating apoptotic response to the level of induced protein (as detected by immunoblotting). We show that 10-fold more mC-D83A-BAK is required to promote similar extent of cell death as mC-V74A-BAK. Similarly, 4-fold more mC-D83A-BAK promotes half of cell death achieved with mC-V74A-BAK (**New Figures 3c, 3d**). The difference in apoptotic response parallels that in liposome permeabilization with full-length V74A and D83A BAK: 4-fold more D83A is required to induce half of liposome permeabilization achieved by V74A (**Revised Figure 2b**).

Besides this issue we believe that our analyses are fully transparent. We present normalized liposome permeabilization traces used to determine the area under the curve (AUC) values and now make a statement in the methods (highlighted on pg. 23) about variability in permeabilization of different liposomes. We have always carefully compared among conditions investigated with the same liposomes and note that although the absolute permeabilization values may differ, the trends for the same conditions are always very similar between different liposomes. When we present liposome permeabilization data we expand the axes so that the normalized traces or AUC bars are best shown. We hope to have addressed this criticism satisfactorily.

3. Mutants that have mixed activities

Indeed, the mutations we tested in this study affect direct activation, autoactivation, and core helix 2-5 dimerization differently (summarized in **New Figure 2d**). We investigated two mutants with mixed activities in detail: L78A (impaired in autoactivation from the ligand side and possibly core helix2-5 dimerization), and I81R (impaired in direct activation, autoactivation from the ligand and receptor side and possibly core helix2-5 dimerization).

In new experiments correlating mitochondrial poration with signature patterns of proteolysis in purified mitochondria (**New Figures 4e**) we reveal that:

- 1) L78A constitutively expressed as pMX-L78A BAK-IRES-GFP in BCL2allKO HCT116 cells (**New Figures 4d, 4e**) is inactive in mitochondrial poration being refractory to direct activation by BID BH3 peptide ligands while exhibiting m-calpain proteolysis patterns that suggest a constitutive open conformation even in the absence of direct activator. We conclude that L78A is a folding mutant that associates with the mitochondria but is inactive in poration. Confirming these observations, we show that Dox-inducible L78A stably expressed as mC-2A-L78A BAK in BCL2allKO HCT116 cells is unable to induce cell death although expressed at levels comparable to Dox-inducible D83A analyzed in response to criticism point 2 above (**New Figure S6e, S6f**).
- 2) Constitutively expressed I81R BAK produced using pMX-BAK-IRES-GFP in BCL2allKO HCT116 cells releases cyt c at high doses of M(3)W(5) BID but not WT BID BH3; I81R BAK exhibits resistance to m-calpain proteolysis in the absence of BH3 activator becoming susceptible to proteolysis in the presence of BH3 activator to generate a stabilized C-bundle fragment; To release cyt c I81R requires high doses of the potent M(3)W(5) BID BH3 activator (>90% occupancy based on K_D) which is consistent with similarly high doses of this peptide required for liposome permeablization by I81R (**New Figure 4e**).
- 3) While effects of mutations downstream asymmetric BH3-in-groove autoactivation have not been evaluated in this manuscript, being the subject of a future study, we confidentially share with reviewer #2 preliminary data on the effect of BAK mutations in the presence of detergent. [Redacted unpublished data]

4. Cellular studies limited by technical challenges (inability to compare with wild type Bak, mutants with cleanest differences in mechanism not evaluable in cells)

In our original submission this reviewer noted that L78A BAK would be ideally suited to test in apoptosis given its predicted impact on autoactivation from the ligand side with minimal effect on direct activation. However, mC-L78A BAK was unstable and quickly degraded in BCL2allKO HCT116 cells. We overcame this challenge by expressing L78A BAK and WT-like V74A BAK as Dox-inducible mC-2A-BAK in BCL2allKO HCT116 cells (2A is the ribosome-skipping peptide) and constitutively using pMX-BAK-IRES-GFP in BCL2allKO HCT116 cells, as detailed in point 3.1) above.

Another technical challenge was expressing WT BAK in BCL2allKO HCT116 cells. We overcame this challenge by using the WT-like V74A mutant. We now show that WT and V74A BAK induce apoptosis similarly when constitutively reconstituted in *bak*^{-/-} *bax*^{-/-} HCT116 cells using pMX-BAK-IRES-GFP retroviral strategy (**New Figures S5e, S5f**), supporting our initial hypothesis that V74A recapitulates the structure-function properties of WT BAK. We therefore used V74A as a surrogate of WT BAK in cells refractory to expression of WT BAK.

REVIEWER 3

The authors have addressed most of the key questions addressed by the reviewers and provided reasonable explanations why some issues cannot be experimentally tested. The

manuscript remains very long and complicated to read, but it contains an impressive amount of work and a relevant take-home message that deserves publication in Nature Communications.

We thank this reviewer for the positive comments. We revised our presentation to improve readability although we have added additional data to satisfy reviewers 1 and 2.

Reviewers' Comments:

Reviewer #2:

Remarks to the Author:

Singh et al report crystal structures of BH3 complexes with BAK, propose a structure-based mechanism that regulates activation (alpha-1 destabilization), and suggest a distinction between relative efficiencies of direct and autoactivation based on mutagenesis analysis and read outs by liposomal release and cellular assays.

1. A remaining concern is over-reliance on liposomal area under the curve (AUC) data that are over-interpreted as showing differences when actual differences do not appear biochemically meaningful. As an example, in Figure 1d, a plot is shown with AUC values ranging from 0 to approximately 12.5. The only difference between the BID (left) and BAK (right) BH3 data is the 0.1 micromolar bar. All other bars (1, 10, 50 micromolar) are essentially identical to one another, yet the EC50s are presented as 1500 +/- 880 for BID BH3 vs. 191 +/- 151 for BAK (this is indicated as micromolar but is likely meant to say nanomolar). Based on these questionable values, the conclusion is drawn that BAK BH3 is a better activator than BID BH3. This type of scenario, including some notable differences between biological replicates, is seen in liposomal data (S3d-e, S4e, 6c). The same issue occurs with cellular data, particularly when value ranges are low (4c). In such cases, statistical significance is unlikely biologically meaningful; for example, a statistically significant difference is recorded for values of ~18 vs. ~22% in I18R Annexin V data (4c bottom row).

2. It is unclear why the authors did not evaluate D83A in the Dox/TRAIL+CHX system, since it could have contrasted nicely with V74A if the results followed the hypothesis (TRAIL+CHX rescue of defective autoactivation), but instead studied a mutant with poor direct activation (I81R) and a mutant that doesn't fold correctly (L78A). The cellular system also seems less reliable or definitive than the authors propose with respect to distinguishing between autoactivation and direct activation pools. Dox treatment causes BAK expression and BAK autoactivation induced death; in this case, BAK is mostly undergoing autoactivation with some unknown or unpredictable quantity of residual inactive BAK available for direct activation. In this context, it would seem that direct activation will always be observed less frequently due to the tonic BAK autoactivation induced by Dox expression. Also, once directly activated by TRAIL+CHX, presumably there is also a component of BAK autoactivation within the pool labeled as direct activation. These issues make the cellular system rather complicated to interpret and not a simple A vs. B quantification as presented.

3. For the part of the story that relies on mutant BID peptides that either do or do not activate BAK depending on the dose, it seemed inconsistent that the crystal structure of the inhibitory W3W5 complex was set up at a 4:1 peptide:protein ratio, yet at 20:1 the peptide is observed to still have an activating function in the liposomal release assay (6c).

4. The manuscript is so dense with subpanels, mutants, and peptide/protein combinations that the reader struggles to decipher the analysis, even though the message of the manuscript could be quite simple, helix-1 destabilization plays a role in BAK activation.

Overall, the shakiness of some of the biochemical and cellular data undermines the conclusions drawn from the otherwise elegant structural work. The nuance that autoactivation is more efficient than direct activation does not seem well supported given some of the subtle numerical differences observed across assays. In addition, Dox-induced expression causes BAK-autoactivation and death, which is not analogous to homeostasis where BAK lies dormant in the mitochondrial membrane until stimulated by activating ligands. Since the identified residues of the helix 1 electrostatic network are not conserved across species (the published mouse BAK structure could be discussed as a way of comparison), the reader is further left wondering about the fundamental importance of the mechanistic finding.

POINT-BY-POINT RESPONSE

Reviewer #2 (Remarks to the Author):

Singh et al report crystal structures of BH3 complexes with BAK, propose a structure-based mechanism that regulates activation (α -1 destabilization), and suggest a distinction between relative efficiencies of direct and autoactivation based on mutagenesis analysis and read outs by liposomal release and cellular assays.

We thank the reviewer for evaluating our revised manuscript. We addressed the latest remarks raised by this reviewer in full as follows.

1. A remaining concern is over-reliance on liposomal area under the curve (AUC) data that are over-interpreted as showing differences when actual differences do not appear biochemically meaningful. As an example, in Figure 1d, a plot is shown with AUC values ranging from 0 to approximately 12.5. The only difference between the BID (left) and BAK (right) BH3 data is the 0.1 micromolar bar. All other bars (1, 10, 50 micromolar) are essentially identical to one another, yet the EC50s are presented as 1500 +/- 880 for BID BH3 vs. 191 +/- 151 for BAK (this is indicated as micromolar but is likely meant to say nanomolar). Based on these questionable values, the conclusion is drawn that BAK BH3 is a better activator than BID BH3. This type of scenario, including some notable differences between biological replicates, is seen in liposomal data (S3d-e, S4e, 6c).

We appreciate the reviewer's points. We have now avoided overinterpreting subtle differences in the data. We had performed the AUC-based EC50 measurement following the previous requests from the other reviewers. While additional data points in the curves would improve estimation of EC50, we have decided to remove mention of EC50 values, except when they are based on clear dose-dependent differences (e.g. V74A vs D83A in liposome permeabilization), and to use another parameter instead, as explained below.

The reviewer points to variations between replicate experiments; we have been transparent about this limitation in our previous response: while measurements can vary considerably, the trends remain consistent. In particular, the lowest dose at which permeabilization is observed is quite robust across experiments: for BAK BH3 (WT or mutants M71A and V74A), permeabilization occurs consistently at 0.1 μ M; for BID BH3, permeabilization is observed from 1 μ M. We propose that might be a better way to identify subtle functional differences among BH3 ligands, and will thus report that measurement instead of EC50 values in Figure 1d, avoiding overinterpretation of the data.

The same issue occurs with cellular data, particularly when value ranges are low (4c). In such cases, statistical significance is unlikely biologically meaningful; for example, a statistically significant difference is recorded for values of ~18 vs. ~22% in I18R Annexin V data (4c bottom row).

We appreciate the criticism and have removed the I81R data, replacing it with new analyses of D83A as described in the next point.

2. It is unclear why the authors did not evaluate D83A in the Dox/TRAIL+CHX system, since it could have contrasted nicely with V74A if the results followed the hypothesis (TRAIL+CHX rescue of defective autoactivation), but instead studied a mutant with poor direct activation (I81R) and a mutant that doesn't fold correctly (L78A).

This is a great suggestion. We now include Dox/TRAIL+CHX analysis of V74A vs D83A in BID-expressing BCL2allKO cells and present the data in new Figures 3e, 3f, (highlighted on pg. 8). Both mutants induce more apoptosis in the presence of TRAIL+CHX suggesting that direct activation stimulated them. The autoactivation impaired D83A BAK mutant induces less apoptosis than V74A under these conditions.

By removing the cellular data for I81R and L78A, the paper has been streamlined and is now less dense.

The cellular system also seems less reliable or definitive than the authors propose with respect to distinguishing between autoactivation and direct activation pools. Dox treatment causes BAK expression and BAK autoactivation induced death; in this case, BAK is mostly undergoing autoactivation with some unknown or unpredictable quantity of residual inactive BAK available for direct activation. In this context, it would seem that direct activation will always be observed less frequently due to the tonic BAK autoactivation induced by Dox expression. Also, once directly activated by TRAIL+CHX, presumably there is also a component of BAK autoactivation within the pool labeled as direct activation. These issues make the cellular system rather complicated to interpret and not a simple A vs. B quantification as presented.

We agree with this reviewer that the contributions of autoactivation and direct activation in cells are more complicated than the A vs B quantification presented previously, and have thus removed that representation from the graphs, allowing the reader to interpret these changes in a more nuanced way in Figure 3.

3. For the part of the story that relies on mutant BID peptides that either do or do not activate BAK depending on the dose, it seemed inconsistent that the crystal structure of the inhibitory W3W5 complex was set up at a 4:1 peptide:protein ratio, yet at 20:1 the peptide is observed to still have an activating function in the liposomal release assay (6c).

We would like to clarify that there is no inconsistency; while the reviewer is correct about the peptide:protein ratios, the concentrations in each setup are vastly different. For the crystal structure, we used 10 mg/mL BAK (0.5 mM) and 2 mM W3W5 peptide; given the K_D of 0.25 μ M between W3W5 BID for BAK, these conditions should yield nearly 99.99% of BAK bound to peptide. For the liposome assay, we used 50 nM BAK and 1 μ M W3W5 BAK, conditions that

should result in 80% BAK bound to peptide; we also expect some dissociation of W3W5 peptide from BAK resulting in activation.

4. The manuscript is so dense with subpanels, mutants, and peptide/protein combinations that the reader struggles to decipher the analysis, even though the message of the manuscript could be quite simple, helix-1 destabilization plays a role in BAK activation.

We agree the manuscript is dense and have now removed cellular data related to I81R and L78A, following the reviewer's comments in point 2. This reduced results and discussion sections, without changing any of our conclusions. We have also streamlined presentation of other data, as explained in response to point 1 and below.

Overall, the shakiness of some of the biochemical and cellular data undermines the conclusions drawn from the otherwise elegant structural work. The nuance that autoactivation is more efficient than direct activation does not seem well supported given some of the subtle numerical differences observed across assays. In addition, Dox-induced expression causes BAK-autoactivation and death, which is not analogous to homeostasis where BAK lies dormant in the mitochondrial membrane until stimulated by activating ligands.

We appreciate the comments and have revised the presentation of the biochemical and cellular data, as detailed in our response to the previous points, to avoid overinterpretation.

Regarding Dox-induced expression not being analogous to homeostasis, we do not claim that in the text and had already addressed this concern in the previous response to another reviewer. Nevertheless, we find that this setup is useful as it allows us to directly investigate autoactivation, and have acknowledged its limitations in the revised text (highlighted on pgs. 12-13).

Since the identified residues of the helix 1 electrostatic network are not conserved across species (the published mouse BAK structure could be discussed as a way of comparison), the reader is further left wondering about the fundamental importance of the mechanistic finding.

The residues of helix 1 electrostatic network are conserved in humans and closely related mammals (Figure 6d), and we have mentioned that in divergent species such as mouse the mechanism will be differently regulated. We do not currently have a structure of mouse BAK bound to a BH3 activator, which would be required to reveal alternative mechanisms of BH3 ligand-induced helix 1 destabilization.

Nonetheless, we now present an overlay of human and mouse apo BAK highlighting helix 1 electrostatic region and explaining similarities and differences (Figure S8c). In mouse BAK, we propose an alternative electrostatic network stabilizing helix 1, comprised of conserved salt bridge E44–R135, buried in a similar location as in human BAK (E46–R137); and solvent-exposed salt bridge Q43–D81, replacing the buried one (R42–D90) in human BAK. We also note a

different arrangement of helix 3 that positions mouse D88 away from its position in human BAK (D90).

We thank again the reviewer for thoroughly evaluating our manuscript. Addressing the points raised by this reviewer has improved our manuscript.

Reviewers' Comments:

Reviewer #2:

None